# An investigation of the thermo-mechanical features of Laohugou Glacier No.12 in Mt. Qilian Shan, western China, using a two-dimensional first-order flowband ice flow model

Yuzhe Wang[1,2], Tong Zhang[3,a], Jiawen Ren[1], Xiang Qin[1], Yushuo Liu[1], Weijun Sun[4], Jizu Chen[1,2], Minghu Ding[3], Wentao Du[1,2], and Dahe Qin[1]

[1]State Key Laboratory of Cryospheric Science, Northwest Institute of Eco-Environment and Resources, Chinese Academy of Sciences, Lanzhou 730000, China
[2]University of Chinese Academy of Sciences, Beijing 100049, China
[3]Insititute of Polar Meteorology, Chinese Academy of Meteorological Sciences, Beijing 100081, China
[4]College of Geography and Environment, Shandong Normal University, Jinan 250014, China
[a]now at: Fluid Dynamics and Solid Mechanics Group, Los Alamos National Laboratory, Los Alamos, New Mexico, 87545, USA

*Correspondence to:* Tong Zhang (zhangtong@cma.gov.cn)

**Abstract.** By combining *in situ* measurements and a two-dimensional thermo-mechanically coupled ice flow model, we investigate the thermo-mechanical features of the largest valley glacier (Laohugou Glacier No.12; LHG12) in Mt. Qilian Shan located in the arid region of western China. Our model results suggest that LHG12, previously considered as fully cold, is probably polythermal, with a lower temperate ice layer overlain by an upper layer of cold ice over a large region of the abla-
5    tion area. Modeled ice surface velocities match well with the *in situ* observations in the east branch (main branch) but clearly underestimate those near the glacier terminus possibly because the convergent flow is ignored and the basal sliding beneath the confluence area is underestimated. The modeled ice temperatures are in very good agreement with the *in situ* measurements from a deep borehole (110 m deep) in the upper ablation area. The model results are sensitive to surface thermal boundary conditions, for example, surface air temperature and near-surface ice temperature. In this study, we use a Dirichlet surface thermal condition constrained by 20 m borehole temperatures and annual surface air temperatures. Like many other alpine
10   glaciers, strain heating is important in controlling the englacial thermal structure of LHG12. Our transient simulations indicate that the accumulation zone becomes colder during the last two decades as a response to the elevated equilibrium-line altitude and the rising summer air temperatures. We suggest that the extent of accumulation basin (the amount of refreezing latent heat from meltwater) of LHG12 has a considerable impact on the englacial thermal status.

## 1 Introduction

The storage of water in glaciers is an important component of the hydrological cycle at different time scales (Jansson et al., 2003; Huss et al., 2010), especially in arid and semi-arid regions such as northwestern China, where many glaciers are currently retreating and disappearing (Yao et al., 2012; Neckel et al., 2014; Tian et al., 2014). Located on the northeastern edge of the Tibetan Plateau (36 – 39 °N, 94 – 104 °E), Mt. Qilian Shan (MQS) develops 2051 glaciers covering an area of approximately

$1057 \text{ km}^2$ with a total ice volume of approximately $51 \text{ km}^3$ (Guo et al., 2014, 2015). Meltwater from MQS glaciers is a very important water resource for the agricultural irrigation and socio-economic development of the oasis cities in northwestern China. Thus, the changes in the MQS glaciers that occur as the climate becomes warmer in the near future are of concern.

Due to logistic difficulties, few MQS glaciers have been investigated in previous decades. However, Laohugou Glacier No.12 (hereafter referred to as LHG12), the largest valley glacier of MQS, has been investigated. Comprised of two branches (east and west), LHG12 is located on the north slope of the western MQS (39°27' N, 96°32' E; Fig. 1), with a length of approximately 10 km, an area of approximately $20 \text{ km}^2$, and an elevation range of 4260 – 5481 m a.s.l. (Liu et al., 2011). LHG12 was first studied by a Chinese expedition during 1958 – 1962 and was considered again in short-term field campaigns in the 1970s and 1980s that were aimed at monitoring glacier changes (Du et al., 2008). Since 2008, the Chinese Academy of Sciences has operated a field station for obtaining meteorological and glaciological measurements of LHG12.

The temperature distribution of a glacier primarily controls the ice rheology, englacial hydrology, and basal sliding conditions (Blatter and Hutter, 1991; Irvine-Fynn et al., 2011; Schäfer et al., 2014). A good understanding of the glacier thermal conditions is important for predicting glacier response to climate change (Wilson et al., 2013; Gilbert et al., 2015), improving glacier hazard analysis (Gilbert et al., 2012), and reconstructing past climate histories (Vincent et al., 2007; Gilbert et al., 2010). Different energy inputs at the glacier surface determine the near-surface ice temperature. In particular, the latent heat released by meltwater refreezing in the percolation zone can largely warm the near-surface area (e.g., Müller, 1976; Huang et al., 1982; Blatter, 1987; Blatter and Kappenberger, 1988; Gilbert et al., 2014b). The importance of surface thermal boundary condition in controlling the thermal regime of a glacier has been recognized by recent thermo-mechanical glacier model studies (e.g., Wilson and Flowers, 2013; Gilbert et al., 2014a; Meierbachtol et al., 2015). Using both *in situ* measurements and numerical models, Meierbachtol et al. (2015) argued that shallow borehole ice temperatures served as better boundary constraints than surface air temperatures in Greenland. However, for the east Rongbuk glacier on Mt. Everest, which is considered polythermal, Zhang et al. (2013) found that the modeled ice temperatures agreed well with the *in situ* shallow borehole observations when using surface air temperatures as the surface thermal boundary condition. Therefore, careful investigation of the upper thermal boundary condition is highly necessary for glaciers in different regions under different climate conditions.

LHG12 is widely considered as an extremely continental type (cold) glacier and is characterized by low temperatures and precipitation (Huang, 1990; Shi and Liu, 2000). However, in recent years, we have observed extensive and widespread meltwater on the ice surface and at the glacier terminus. In addition, percolation of snow meltwater consistently occurs in the accumulation basin during the summer. Therefore, we address the following two pressing questions in this study: (1) What is the present thermal status of LHG12? and (2) How do different surface thermal boundary conditions impact the modeled ice temperature and flow fields? Because temperate ice can assist basal motion and accelerate glacier retreat, understanding the current thermal status of LHG12 is very important for predicting its future dynamic behaviour.

To answer these questions, we conduct diagnostic and transient simulations for LHG12 by using a thermo-mechanically coupled first-order flowband ice flow model. This paper is organized as follows: First, we provide a detailed description of the glaciological datasets of LHG12. Then, we briefly review the numerical ice flow model and the surface mass balance model used in this study. Next, we perform both diagnostic and transient simulations. In the diagnostic simulations, we first investigate

the sensitivities of ice flow model parameters, and we then explore the impacts of different surface thermal boundary conditions and assess the contributions of heat advection, strain heating, and basal sliding to the temperature field of LHG12. Transient simulations are performed during the period 1961–2011. We then compare the transient results with measured ice surface velocities and ice temperature profile obtained from a deep borehole. We also investigate the evolution of the temperature profile in the deep borehole and the impacts of initial equilibrium line altitude (ELA) on the thermal regime of the glacier. Finally, we discuss the limitations of our model and present the important conclusions that resulted from this study.

## 2 Data

Most *in situ* observations, e.g., borehole ice temperatures, surface air temperatures and ice surface velocities, have been made on the east branch (main branch) of LHG12 (Fig. 1). Measurements on the west branch are sparse and temporally discontinuous. Thus, we only consider the *in situ* data from the east tributary when building our numerical ice flow model.

### 2.1 Glacier geometry

In July – August 2009 and 2014, two ground-penetrating radar (GPR) surveys were conducted on LHG12 using a pulseEKKO PRO system with center frequencies of 100 MHz (2009) and 50 MHz (2014) (Fig. 1b). Wang et al. (2016) have presented details regarding the GPR data collection and post-processing.

As shown in Fig. 2, the east branch of LHG12 has a mean ice thickness of approximately 190 m. We observed the thickest ice layer (approximately 261 m) at 4864 m a.s.l. Generally, the ice surface of LHG12 is gently undulating, with a mean slope of 0.08°, and the bed of LHG12 shows significant overdeepening in the middle of the center flowline (CL) (Fig. 2). To account for the lateral effects exerted by glacier valley walls in our 2D ice flow model, we parameterize the lateral drag using the glacier half widths. Based on the GPR measurements on LHG12, we parameterize the glacier cross-sections by a power law function $z = aW(z)^b$, where $z$ and $W(z)$ are the vertical and horizontal distances from the lowest point of the profile, and $a$ and $b$ are constants representing the flatness and steepness of the glacier valley, respectively (Svensson, 1959). The $b$ values for LHG12 range from 0.8 to 1.6, indicating that the valley is approximately "V"-shaped (Wang et al., 2016). As an input for the flowband ice flow model, the glacier width, $W$, was also calculated by ignoring all tributaries (including the west branch) (Fig. 1b and 2).

### 2.2 Ice surface velocities

The ice surface velocities of LHG12 were determined from repeated surveys of stakes drilled into the ice surface. All stakes were located in the distance between km 0.6 – 7.9 along the CL (Fig. 3), spanning an elevation range of 4355 – 4990 m.a.s.l. (Fig. 1b). We measured the stake positions using a real-time kinematic (RTK) fixed solution by a South Lingrui S82 GPS system (Liu et al., 2011). The accuracy of the GPS positioning is on the order of a few centimeters and the uncertainty of the calculated ice surface velocities is estimated to be less than 1 m a$^{-1}$. Because it is difficult to conduct fieldwork on LHG12 (due to, e.g., crevasses and supra-glacial streams), it was nearly impossible to measure all stakes each observational year. Thus, the

current dataset includes annual ice surface velocities from 2008 – 2009 and 2009 – 2010, summer measurements from June 17 – August 30, 2008, and winter measurements from February 1 – May 28, 2010.

The *in situ* ice surface velocities shown in Fig. 3 are all from stakes near the CL (Fig. 1b). Small ice surface velocities ($<$ 17 m $a^{-1}$) are clearly visible in the upper accumulation (km 0 – 1.2) and lower ablation areas (km 6.5 – 9.0) (Fig. 3). Fast ice flow ($>$ 30 m $a^{-1}$) can be observed between elevations of 4700 – 4775 m a.s.l. (km 4.0 – 5.0), where the ice surface velocities during the summer are approximately 6 m $a^{-1}$ greater than the annual mean velocity ($<$ 40 m $a^{-1}$). Measurements of winter ice surface velocities ($<$ 10 m $a^{-1}$) are only available near the glacier terminus showing a clear inter-annual variation of the ice flow speeds.

## 2.3 Borehole ice temperature

In August 2009 and 2010, we drilled three 25 m deep shallow boreholes on LHG12 (Fig. 1b). One borehole was drilled in the upper ablation area (site 2, approximately 4900 m a.s.l.) and two boreholes were drilled at the AWS locations (sites 1 and 3). The snow/ice temperatures were measured in the boreholes during the period of October 1, 2010 – September 30, 2011. The seasonal variations of the snow/ice temperatures in the shallow boreholes are presented in Figs. 4a, b and c. Our measurements show very little fluctuation ($\pm 0.4\,^\circ$C) in the ice temperatures over the depth range of 20 – 25 m. Below the 3 m depth, the annual mean temperature profiles for sites 1 and 2 show a linearly increase in temperature with depth, while the annual mean temperature profile for site 3 is convex upward. The mean annual 20 m ice temperatures ($T_{20m}$) at sites 1, 2, and 3 are $5.5\,^\circ$C, $3.0\,^\circ$C, and $9.5\,^\circ$C higher than the mean annual air temperatures ($T_{air}$), respectively. Despite its higher elevation, the near-surface snow/ice temperatures below a depth of 5 m at site 3 are greater than those in the ablation area (sites 1 and 2), largely due to the latent heat released as the meltwater entrapped in the surface snow layers refreezes.

To determine the englacial thermal conditions of LHG12, we drilled an ice core to a depth of 167 m in the upper ablation area of LHG12 (4971 m a.s.l., Fig. 1b). In October 2011, the ice temperature were measured to a depth of approximately 110 m using a thermistor string after 20 days of the drilling, as shown in Fig. 4d. The string consists of 50 temperature sensors with a vertical spacing of 0.5 m and 10 m at the ice depths of 0 – 20 m and 20 – 110 m, respectively. The accuracy of the temperature sensor is around $\pm 0.05\,^\circ$C (Liu et al., 2009). From Fig. 4d we can see that the temperature profile is close to linear with a temperature gradient of around $0.1\,^\circ$C m$^{-1}$ at the depths of 9 – 30 m. Below the depth of 30 m, the ice temperature demonstrates a linear relationship with depth as well but with a smaller temperature gradient of around $0.034\,^\circ$C m$^{-1}$.

## 2.4 Meteorological data

Two automatic weather stations (AWS) were deployed on LHG12, one in the ablation area at 4550 m a.s.l. (site 1, Fig. 1b), and the other in the accumulation area at 5040 m a.s.l. (site 3). During the period of 2010 – 2013, the mean annual air temperatures (2 m above the ice surface) at sites 1 and 3 were $-9.2\,^\circ$C and $-12.2\,^\circ$C, respectively, indicating a lapse rate of $-0.0061\,^\circ$C m$^{-1}$.

The historical knowledge of the surface air temperature and the ELA of LHG12 is a necessity for running the transient model. We reconstruct the past air temperature on LHG12 based on the observations of the Tuole meteorological station (3367 m a.s.l.), which is approximately 175 km southeast to LHG12. We get the precipitation on LHG12 by downscaling the CAPD

(China Alpine Region Month Precipitation Dataset) in Qilian Shan. CAPD provides the monthly sum of precipitation during the period of 1960 – 2013 with a grid spacing of 1 km. We calculate the precipitation on LHG12 from its surrounding 91 grids in CAPD based on the relationship between precipitation and geometric parameters. More details about the reconstruction of both air temperature and precipitation for LHG12 can be found in the Supplement.

## 5    3    Thermo-mechanical ice flow model

In this study, we use the same two-dimensional (2D), thermo-mechanically coupled, first-order, flowband ice flow model as in Zhang et al. (2013). Therefore, we only address a very brief review of the model here.

### 3.1    Ice flow model

We define $x$, $y$, and $z$ as the horizontal along-flow, horizontal across-flow and vertical coordinates, respectively. By assuming
the vertical normal stress as hydrostatic and neglecting the bridging effects (Blatter, 1995; Pattyn, 2002), the equation for momentum balance is given as

$$\frac{\partial}{\partial x}(2\sigma'_{xx} + \sigma'_{yy}) + \frac{\partial \sigma'_{xy}}{\partial y} + \frac{\partial \sigma'_{xz}}{\partial z} = \rho g \frac{\partial s}{\partial x}, \tag{1}$$

where $\sigma'_{ij}$ is the deviatoric stress tensor, $\rho$ is the ice density, $g$ is the gravitational acceleration, and $s$ is the ice surface elevation. The parameters used in this study are given in Table 1.

The constitutive relationship of ice dynamics is described by the Glen's flow law (Cuffey and Paterson, 2010, p. 60–61)

$$\sigma'_{ij} = 2\eta \dot{\epsilon}_{ij}, \quad \eta = \frac{1}{2}A^{-1/n}(\dot{\epsilon}_e + \dot{\epsilon}_0)^{(1-n)/n}, \tag{2}$$

where $\eta$ is the ice viscosity, $\dot{\epsilon}_{ij}$ is the strain rate, $n$ is the flow law exponent, $A$ is the flow rate factor, $\dot{\epsilon}_e$ is the effective strain rate, and $\dot{\epsilon}_0$ is a small number used to avoid singularity. The flow rate factor is parameterized using the Arrhenius relationship as

$$A(T) = A_0 \exp(-\frac{Q}{RT}), \tag{3}$$

where $A_0$ is the pre-exponential constant, $Q$ is the activation energy for creep, $R$ is the universal gas constant, and $T$ is the ice temperature. The effective strain rate $\dot{\epsilon}_e$ is related to the velocity gradient by

$$\dot{\epsilon}_e^2 \simeq \left(\frac{\partial u}{\partial x}\right)^2 + \left(\frac{\partial v}{\partial y}\right)^2 + \frac{\partial u}{\partial x}\frac{\partial v}{\partial y} + \frac{1}{4}\left(\frac{\partial u}{\partial y}\right)^2 + \frac{1}{4}\left(\frac{\partial u}{\partial z}\right)^2, \tag{4}$$

where $u$ and $v$ are the velocity components along the $x$ and $y$ direction, respectively. By assuming $\partial v/\partial y = (u/W)(\partial W/\partial x)$,
we parameterize the lateral drag, $\sigma'_{xy}$, as a function of the flowband half width, $W$, following Flowers et al. (2011)

$$\sigma'_{xy} = -\frac{\eta u}{W}. \tag{5}$$

For an easy numerical implementation, we reformulate the momentum balance equation (1) as

$$\frac{u}{W}\left\{2\frac{\partial\eta}{\partial x}\frac{\partial W}{\partial x}+2\eta\left[\frac{\partial^2 W}{\partial x^2}-\frac{1}{W}\left(\frac{\partial W}{\partial x}\right)^2\right]-\frac{\eta}{W}\right\}$$

$$+\frac{\partial u}{\partial x}\left(4\frac{\partial\eta}{\partial x}+\frac{2\eta}{W}\frac{\partial W}{\partial x}\right)+\frac{\partial u}{\partial z}\frac{\partial\eta}{\partial z}+4\eta\frac{\partial^2 u}{\partial x^2}+\eta\frac{\partial^2 u}{\partial z^2}=\rho g\frac{\partial s}{\partial x}, \tag{6}$$

where the ice viscosity is expressed as

$$\eta=\frac{1}{2}A^{-1/n}\left[\left(\frac{\partial u}{\partial x}\right)^2+\left(\frac{u}{W}\frac{\partial W}{\partial x}\right)^2+\frac{u}{W}\frac{\partial u}{\partial x}\frac{\partial W}{\partial x}+\frac{1}{4}\left(\frac{\partial u}{\partial z}\right)^2+\frac{1}{4}\left(\frac{u}{W}\right)^2+\dot{\epsilon_0}^2\right]^{(1-n)/2n}. \tag{7}$$

## 3.2 Ice temperature model

The ice temperature field can be calculated using a 2D heat transfer equation (Pattyn, 2002),

$$\frac{\partial T}{\partial t}=\frac{k}{\rho c_p}\left(\frac{\partial^2 T}{\partial x^2}+\frac{\partial^2 T}{\partial z^2}\right)-\left(u\frac{\partial T}{\partial x}+w\frac{\partial T}{\partial z}\right)+\frac{4\eta\dot{\epsilon}_e^2}{\rho c_p}, \tag{8}$$

where $w$ is the vertical ice velocity, $k$ and $c_p$ are the thermal conductivity and heat capacity of the ice, respectively.

The pressure melting point of the ice, $T_{\mathrm{pmp}}$, is described by the Clausius-Clapeyron relationship

$$T_{\mathrm{pmp}}=T_0-\beta(s-z), \tag{9}$$

where $T_0$ is the triple-point temperature of water and $\beta$ is the Clausius-Clapeyron constant. Following Zhang et al. (2013), we determined the position of the cold-temperate ice transition surface (CTS) by considering the following two cases: (1) melting condition, i.e., cold ice flows downward into the temperate ice zone, and (2) freezing condition, i.e., temperate ice flows upward into the cold ice zone (Blatter and Hutter, 1991; Blatter and Greve, 2015). For the melting case, the ice temperature profile at the CTS simply follows a Clausius-Clapeyron gradient ($\beta$). However, for the freezing case, the latent heat, $Q_r$, that is released when the water contained in the temperate zone refreezes is determined as (Funk et al., 1994)

$$Q_r=\mu w\rho_w L, \tag{10}$$

where $\mu$ is the fractional water content of the temperate ice, $\rho_w$ is the water density and $L$ is the latent heat of freezing. In this case, following Funk et al. (1994), the ice temperature gradient at the CTS can be described as

$$\frac{\partial T}{\partial z}=-\frac{Q_r}{k}+\beta. \tag{11}$$

## 3.3 Free surface

The free surface evolution follows the kinematic boundary equation,

$$\frac{\partial s}{\partial t}=b_n+w_s-u_s\frac{\partial s}{\partial x}, \tag{12}$$

where $s(x,t)$ is the free surface elevation, $u_s$ and $w_s$ are the surface velocity components in $x$ and $z$, and $b_n$ is the surface mass balance.

### 3.4 Boundary conditions

#### 3.4.1 Boundary conditions for ice flow model

For the ice flow model, we assume a stress-free condition for the glacier surface, and use the Coulomb friction law to describe the ice-bedrock interface where the ice slips (Schoof, 2005),

$$\tau_b = \Gamma \left( \frac{u_b}{u_b + \Gamma^n N^n \Lambda} \right)^{1/n} N, \qquad \Lambda = \frac{\lambda_{\max} A}{m_{\max}}, \tag{13}$$

where $\tau_b$ and $u_b$ are the basal drag and velocity, respectively, $N$ is the basal effective pressure, $\lambda_{\max}$ is the dominant wavelength of the bed bumps, $m_{\max}$ is the maximum slope of the bed bumps, and $\Gamma$ and $\Lambda$ are geometrical parameters (Gagliardini et al., 2007). Here we take $\Gamma = 0.84 m_{\max}$ following Gagliardini et al. (2007) and Flowers et al. (2011). The basal effective pressure in the friction law, $N$, is defined as the difference between the ice overburden pressure and the basal water pressure (Gagliardini et al., 2007; Flowers et al., 2011),

$$N = \rho g H - P_w = \phi \rho g H, \tag{14}$$

where $H$ and $P_w$ are the ice thickness and basal water pressure, respectively, and $\phi$ implies the ratio of basal effective pressure to the ice overburden pressure. The basal drag is defined as the sum of all resistive forces (Van der Veen and Whillans, 1989; Pattyn, 2002). It should be noted that the basal sliding is only permitted when basal ice temperature reaches the local pressure-melting point.

#### 3.4.2 Boundary conditions for ice temperature model

We apply a Dirichlet temperature constraint ($T_{\mathrm{sbc}}$) on the ice surface in the temperature model. In some studies, $T_{\mathrm{sbc}} = T_{\mathrm{air}}$ is used (e.g. Zhang et al., 2013), which, as suggested by recent studies, could result in cold bias in ice temperature simulations (Meierbachtol et al., 2015). By contrast, Meierbachtol et al. (2015) recommended using the near-surface temperature $T_{\mathrm{dep}}$ at depth where inter-annual variations of air temperatures are damped (15 – 20 m deep) as a proxy for the annual mean ice surface temperature. One advantage of using $T_{\mathrm{dep}}$ is that the effects of refreezing meltwater and the thermal insulation of winter snow can be included in the model (Huang et al., 1982; Blatter, 1987). In fact, the condition $T_{\mathrm{dep}} = T_{\mathrm{air}}$ is acceptable only in dry and cold snow zones (Cuffey and Paterson, 2010, p. 403–404); however, $T_{\mathrm{dep}} > T_{\mathrm{air}}$ is often observed in zones where meltwater is refreezing in glaciers, such as the LHG12 (Fig. 4). In this study, we set $T_{\mathrm{sbc}}$ in the accumulation zone to the glacier near-surface temperature $T_{\mathrm{dep}}$, while $T_{\mathrm{sbc}}$ in the ablation area is prescribed by a simple parameterization (Lüthi and Funk, 2001; Gilbert et al., 2010) as

$$T_{\mathrm{sbc}} = \begin{cases} T_{\mathrm{dep}}, & s > ELA, \\ T_{\mathrm{air}} + c, & s \leq ELA, \end{cases} \tag{15}$$

where ELA is the equilibrium-line altitude, and $c$ is a tuning parameter implicitly accounting for effects of the surface energy budget and the lapse rate of air temperature (Gilbert et al., 2010). We use Eq. (15) as the surface thermal boundary condition

of the reference experiment after comparing another numerical experiment by setting $T_{\text{sbc}} = T_{\text{air}}$ (E-air) (see Sect. 6.1.2 for details).

At the ice-bedrock interface, we apply the following Neumann-type boundary condition in the temperature model,

$$\frac{\partial T}{\partial z} = -\frac{G}{k}, \tag{16}$$

where $G$ is the geothermal heat flux. We here use a constant geothermal heat flux, 40 mW m$^{-2}$, an *in situ* measurement from the Dunde ice cap in the western MQS (Huang, 1999), over the entire model domain.

## 3.5 Numerical solution

In our model we use a finite difference discretization method and a terrain-following coordinate transformation. The numerical mesh we use contains 61 grid points in $x$ and 41 layers in $z$. The ice flow model (Eq. (6)) is discretized with a second-order centered difference scheme while the ice temperature model (Eq. (8)) employs a first-order upstream difference scheme for the horizontal heat advection term and a node-centered difference scheme for the vertical heat advection term and the heat diffusion terms. The velocity and temperature fields are iteratively solved by a relaxed Picard subspace iteration scheme (De Smedt et al., 2010) in Matlab. The free surface evolution (Eq. (12)) is solved using a Crank-Nicolson scheme. More details are given in Zhang et al. (2013).

## 4 Surface mass balance model

In our parameterization of the surface thermal boundary condition, a transient ELA is important in controlling the extent of accumulation zone which can be largely warmed by the refreezing of meltwater. We estimate the annual surface mass balance $b_n$ of LHG12 during 1960–2013, and determine the position of ELA by $b_n = 0$.

The daily ablation at elevation $z$, $m_a(z)$, is calculated based on a degree-day method (Braithwaite and Zhang, 2000):

$$m_a(z) = f_m \text{PDD}(z), \tag{17}$$

where $f_m$ is the degree-day factor, and PDD is the daily positive degree-day sum for glacier surface melt,

$$\text{PDD} = T_{\text{mean}} - T_m, \tag{18}$$

where $T_{\text{mean}}$ is the daily mean air temperature, and $T_m$ is the threshold temperature when melt occurs. Note that surface melt may happen even if $T_{\text{mean}} < 0\,^{\circ}\text{C}$, due to the positive air temperature during the day, indicating a negative $T_m$ in such cases.

As the CAPD dataset only provides monthly precipitation sums, we estimate the daily precipitation on LHG12 by assuming a uniformly distributed precipitation over a month. The daily accumulation at elevation $z$, $m_c(z)$, is calculated as follows,

$$m_c(z) = \begin{cases} f_P \cdot P_{\text{total}}, & T_{\text{mean}} < T_{\text{crit1}}, \\ f_P \cdot \frac{T_{\text{crit2}} - T_{\text{mean}}}{T_{\text{crit2}} - T_{\text{crit1}}} \cdot P_{\text{total}}, & T_{\text{crit1}} \leq T_{\text{mean}} \leq T_{\text{crit2}}, \\ 0, & T_{\text{mean}} > T_{\text{crit2}}, \end{cases} \tag{19}$$

where $P_{total}$ is the daily precipitation, $T_{crit1}$ and $T_{crit2}$ are the threshold temperatures for the snow and rain transition, and $f_P$ is a tuning parameter for the precipitation to account for the uncertainties of the gridded CAPD data and the downscaling method. The model is well calibrated by investigating the sensitivities of model parameters and by comparing the simulated results with the observed mass balance during 2010–12 (see Figs. S5–8 in the Supplement).

## 5   Simulation strategies

### 5.1   Surface relaxation

In order to remove the uncertainties remained in the model initial conditions (including initial topography and model parameters) (Zwinger and Moore, 2009; Seroussi et al., 2013), we allow the free surface to relax for a period of 3 years under a constant present-day surface mass balance and zero basal sliding (see details in the Supplement). The time step for the relaxation experiment is set to 0.1 year. We apply surface relaxation before running all of the diagnostic and transient simulations in this study.

### 5.2   Diagnostic simulations

First, we explore the model sensitivities to geometrical bed parameters ($\lambda_{max}$ and $m_{max}$), ratio of basal effective pressure ($\phi$), water content ($\mu$), geothermal heat flux ($G$), and valley shape index ($b$) using diagnostic simulations with relaxed present-day geometry of LHG12. Then, we calibrate the parameters in surface thermal boundary condition (ELA$_0$, $T_{dep}$ and $c$), where ELA$_0$ is the initial ELA, by first running a diagnostic simulation with a given set of parameters (ELA$_0$, $T_{dep}$ and $c$) and then performing a transient simulation using the diagnostic run as an initial condition with time-dependent ELA and surface air temperature (see Sect. 5.3). This numerical process of finding the optimal parameter set of ELA$_0$, $T_{dep}$ and $c$ is repeated until we find a good agreement between the modeled and measured deep borehole temperatures in 2011. Finally, we perform experiments to investigate the sensitivities of the thermo-mechanical model to different surface thermal boundary conditions (E-ref-D and E-air). We also perform four experiments (E-advZ, E-advX, E-strain, and E-slip) to explore the effects of heat advection, strain heating, and basal sliding on the thermal distribution and flow dynamic behaviours of LHG12.

### 5.3   Transient simulations

To investigate the impacts of historical climate conditions on the thermal regime of LHG12, time-dependent simulations are performed from 1961 to 2011. The diagnostic run with the calibrated surface thermal parameters is used as the initial condition of the transient simulations. In our simulations, we assume that the surface temperature ($T_{dep}$) in the accumulation zone is constant in time and space. Due to a lack of *in situ* observations (e.g. firn thickness, firn densities) and coupled heat-water transfer model (e.g. Gilbert, 2012; Wilson, 2013), we do not simulate the complex processes of heat exchange in the accumulation zone. To understand how the thermal status varies over time in the deep borehole, we design three transient simulations (E-cold, E-warm, E-ref-T) by setting $T_{dep}$ to $-5\,°C$, $-1\,°C$ and $-1.8\,°C$ (as calibrated in our diagnostic simulations; see Sect.

6.1.1), respectively. We also design two experiments (E-high and E-low) to explore the impacts of the initial ELA ($\text{ELA}_0$) on the thermal regime of LHG12.

In the transient model, the mean annual air temperature ($T_{\text{air}}$) and ELA are allowed to vary in time. However, we keep the glacier geometry fixed in the transient simulations for two main reasons: (1) The tributaries of LHG12, which our flowband model neglects, may have non-negligible inflow ice fluxes that impact the mass continuity equation; (2) The mean surface elevation change above 4600 m a.s.l. (the confluence area) over 1957–2014 is close to negligible (approximately $-10.4$ m, around 4.4% of the mean ice thickness) (see Fig. S10 in the Supplement).

## 6  Simulation results and discussions

### 6.1  Diagnostic simulations

#### 6.1.1  Parameter sensitivity

As shown in Figs. 5 and 6, we conduct a series of sensitivity experiments to investigate the relative importance of different model parameters ($\lambda_{\text{max}}$, $m_{\text{max}}$, $\phi$, $\mu$, $G$, $b$) on ice flow speeds and temperate ice zone (TIZ) sizes by varying the value of one parameter while holding the others fixed. We set ELA to 4980 m a.s.l., as observed in the 2011 Landsat image of LHG12. $T_{\text{dep}}$ is set to $-2.7\,^{\circ}$C which is the 20 m deep ice temperature at site 3. $c$ is set to $-4.3\,^{\circ}$C, which is the mean differences between the 20 m deep ice temperatures and mean annual air temperatures at sites 1 and 2.

The friction law parameters, $\lambda_{\text{max}}$ and $m_{\text{max}}$, which describe the geometries of bedrock obstacles (Gagliardini et al., 2007; Flowers et al., 2011), have non-negligible impacts on the model results. As shown in Figs. 5a and b, the modeled velocities and TIZ sizes increase as $\lambda_{\text{max}}$ increases and $m_{\text{max}}$ decreases, similar to the results observed by Flowers et al. (2011) and Zhang et al. (2013). A large increase in the modeled basal sliding velocity occurs when $m_{\text{max}}$ is lower than 0.2. The ratio, $\phi$, is an insensitive parameter in our model when it is larger than 0.3 (Fig. 5c). Although the water content, $\mu$, in the ice does not directly impact the ice velocity simulations (the flow rate factor $A$ is assumed independent of the water content in ice), it can affect the temperature field, and consequently influence $A$ and the ice velocities (Fig. 5d). Fig. 6d shows that increasing the water content may result in larger TIZ size. In addition, we test the sensitivity of the model to different geothermal heat flux values. A larger geothermal heat flux can result in a larger TIZ but has a limited impact on modeled ice velocity (Figs. 5e and 6e). As shown by Zhang et al. (2013), our model results are mainly controlled by the shape of the glacial valley, specifically the $b$ index (Sect. 2.1). A large value of $b$ indicates a flat glacial valley and suggests that a small lateral drag is exerted on the ice flow (Figs. 5f and 6f).

Based on the sensitivity experiments described above, we adopt a parameter set of $\lambda_{\text{max}} = 4$ m, $m_{\text{max}} = 0.3$, $\phi = 1$ (no basal water pressure), $\mu = 3\%$, $G = 40$ mW m$^{-2}$, and $b = 1.2$ as a diagnostic reference in our modeling experiment (E-ref-D). To calibrate the surface thermal parameters, we perform a series of model runs by varying $\text{ELA}_0$, $T_{\text{dep}}$ and $c$ in the range of 4940–5010 m a.s.l., $-3.3$–$-1.5\,^{\circ}$C and 1–6$\,^{\circ}$C, respectively. The performance of different parameter combinations is evaluated by comparing the root-mean-squares (RMS) of the differences between the measured and modeled temperatures below the 40

m depth of the deep borehole (Fig. 7). We find the RMSs are sufficiently small ($< 0.11\,°C$) when $ELA_0 = 4940$ m a.s.l., $T_{dep} = -1.8\,°C$ and $c = 3\,°C$ (Fig. 7a–f and h). The sensitivity experiments of $ELA_0$ and their impacts on the thermal regime of LHG12 are discussed in Sect. 6.2.3.

### 6.1.2 Choice of surface thermal boundary condition

To investigate the impacts of different surface thermal boundary conditions on the thermo-mechanical fields of LHG12, we perform an experiment (E-air) in which we set $T_{sbc} = T_{air}$, and compare its results with those of E-ref-D. Fig. 8a shows that the ice temperatures along the CL are highly sensitive to $T_{sbc}$. From E-air, it is observed that LHG12 becomes fully cold (Fig. 8b), with an average field temperature $6.3\,°C$ lower than that of E-ref-D (Fig. 8a), which decreases the ice surface velocity by approximately $11$ m a$^{-1}$ (Fig. 8c).

As noted above, the dynamics of LHG12 can be strongly influenced by the choices of different surface thermal boundary conditions. For LHG12, most accumulation and ablation events overlap in summer (Sun et al., 2012). The meltwater entrapped in snow and moulins can refreeze at a depth where the temperature is below the melting point. Refreezing of meltwater releases large amounts of latent heat, which can significantly raise the near-surface snow/ice temperatures and result in the warm bias of $T_{20m}$ compared to $T_{air}$ (Fig. 4). Therefore, compared with E-air, the experiment E-ref-D better incorporates the effects of meltwater refreezing in the accumulation basin into the prescribed surface thermal boundary constraints resulting in more accurate simulations of ice temperature and flow fields.

### 6.1.3 Roles of heat advection, strain heating and basal sliding

To assess the relative contributions of heat advection and strain heating to the thermo-mechanical field of LHG12, we conduct three experiments (E-advZ, E-advX, and E-strain), in which the vertical advection, horizontal along-flow advection and strain heating were neglected, respectively. In addition, to investigate the effects of basal sliding predicted by the Coulomb friction law on the thermal state and flow dynamics of LHG12, we perform an experiment (E-slip) with $u_b = 0$.

Fig. 8 compares the ice velocity and temperature results of E-advZ, E-advX, E-strain and E-slip with those of E-ref-D. If the vertical advection is neglected (E-advZ), cold ice at the glacier surface cannot be transported downwards into the interior of LHG12, and therefore LHG12 becomes warmer (Fig. 8a and b) and flows faster relative to other experiments (Fig. 8c). Fig. 8a shows a transition of the mean column ice temperature at km 1.8 (the horizontal position of ELA) in E-advX. This transition arises from the discontinuous surface thermal boundary conditions across ELA. Compared with the reference experiment, E-advX predicts lower field temperatures (by $2.3\,°C$ on average, Fig. 8a) and much smaller surface ice velocities (by $7.2$ m a$^{-1}$ on average, Fig. 8c). Because the accumulation basin of LHG12 is relatively warm, E-advX, which neglects the horizontal transport of ice from upstream to downstream, predicts much colder conditions for LHG12 and the modeled temperate ice only appears in the accumulation zone (Fig. 8d). We observe that strain heating contributes greatly to the thermal configuration of LHG12. If we leave away the strain heating, the modeled mean ice temperature field is lower than that of E-ref-D by $0.4\,°C$ on average (Fig. 8a). The TIZ becomes much thinner compared to E-ref-D, indicating the importance of strain heating in the formation of basal temperate ice (Fig. 8d). Previous studies have suggested that basal sliding can significantly influence the

thermal structures and velocity fields of glaciers (e.g. Wilson et al., 2013; Zhang et al., 2015). However, in this study the neglect of basal sliding (E-slip) results in a temperature field very similar to that of E-ref-D. We attribute this difference to the relatively small modeled basal sliding values for LHG12.

## 6.2 Transient simulations

### 6.2.1 Comparison with *in situ* observations

In the reference experiment (E-ref-T), we simulate the distributions of horizontal ice velocities and temperatures in a transient manner (Figs. 9a and c). Next, the model results are compared to the measured ice surface velocities and the ice temperature profile in the deep borehole (Figs. 9b and d). Generally, the modeled ice surface velocities are in good agreement with *in situ* observations from the glacier head to km 4.8 along the CL (Fig. 9b). However, from km 4.8 to the glacier terminus, our model generally underestimates the ice surface velocities as shown in all simulations in Fig. 5. There are three possible reasons for this underestimation. First, the model neglects the convergent ice fluxes from the west branch. Second, an enhanced basal sliding zone may exist at the confluence area, which is not captured by the model. Third, the transient model uses a fixed topography, which may fail to capture the time-dependent glacier changes that can largely influence the ice flow dynamics. Here we verify the first and second hypotheses by conducting two other experiments, i.e. E-W and E-WS. In E-W the glacier widths are increased by 450 m at km 5.8 – 7.3 as a proxy of including the impact of the convergent flow from the west branch (Fig. 10). In E-WS, except for the same glacier width increasing as in E-W, we also increase $\lambda_{\max}$ by 100% and decrease $m_{\max}$ by 33% for accelerating the basal sliding at km 5.8 – 7.3 (Fig. 10). We can clearly find that while both factors have a non-negligible contribution to the model results, the basal sliding may play a bit more important role in the confluence area. This indicates a need of considering glacier flow branches and spatially variable sliding law parameters in real glacier modeling studies.

The model predicts a TIZ overlain by cold ice over a horizontal distance of km 0.6 – 6.5 (Fig. 9c). In addition, we further compare our model results with the *in situ* 110 m deep ice temperature measurements (Fig. 9d). Modeled and measured borehole temperature profiles show a close match within a root-mean-square difference of $0.2\,^{\circ}$C below the 20 m depth. Because *in situ* ice temperature data below the 110 m depth have not been obtained, we are unable to compare the modeled and measured ice temperatures at the ice-bedrock interface.

### 6.2.2 Evolution of the borehole temperature profile

The transient temperature profiles in our reference experiment (E-ref-T) show a cooling trend in the upper surface (0–50 m deep) particularly during the period 1991–2011 (Fig. 11a). This surface cooling trend is also suggested by the experiments E-cold and E-warm in which the thermal parameter $T_{\mathrm{dep}}$ is set to $-5\,^{\circ}$C and $-1\,^{\circ}$C, respectively. However, E-cold (E-warm) underestimates (overestimates) the ice temperature in the deep borehole, compared with the observation. From temperature profile modeled by the reference diagnostic experiment (E-ref-D; Fig. 11a), we can see that the shape of the upper temperature profile (0–30 m) is still difficult to simulate.

In the surface thermal boundary condition, the parameter $T_{\text{dep}}$ simply represent a mean thermal status of the near-surface region in the accumulation zone. The reference transient experiment, which adopts the tuned value of $T_{\text{dep}}$ ($-1.8\,^\circ$C), shows a good agreement between the modeled and observed temperature profile, suggesting a possibly supportive evidence that the calibrated $T_{\text{dep}}$ may indeed contain part of the historical climate information of LHG12. In Figure 11b, we can see that both

the summer (June, July and August) air temperature and ELA appear a slight (large) increase during 1971–1991 (1991–2011), which can explain the small (large) decrease of ice temperatures for all model experiments over the same time period, indicating that LHG12 (or similar type glaciers) may not accordingly become cold under a cold climate scenario, when a relatively large accumulation basin grows and more refreezing latent heat from meltwater is released.

### 6.2.3 Impacts of the initial ELA on the glacier thermal regime

For LHG12, ELA determines the size of accumulation zone and the amount of latent heat released by meltwater refreezing. In the above transient simulations (see Sect. 6.2.1 and 6.2.2), the initial conditions are generated by diagnostic runs in which the $\text{ELA}_0$ is set to 4940 m a.s.l. To investigate the impacts of initial ELA on the thermal regime of LHG12, we perform two additional experiments with different initial ELAs, i.e. E-high ($\text{ELA}_0 = 4800$ m a.s.l.) and E-low ($\text{ELA}_0 = 5000$ m a.s.l.).

The initial differences of the column mean temperature between the sensitivity experiments (E-high and E-low) and the

15 reference experiment (E-ref-T) are significant near the downstream of the $\text{ELA}_0$, as can be seen in Fig. 12. After 50-year runs, however, they are largely reduced because we set the same surface thermal boundary conditions for E-high/E-low and E-ref-T during the transient simulations. But we can still clearly observe two phase shifts of temperature difference in space between the model initialization and the year 2011. By heat advection the temperature field of the downstream and the deep part of LHG12 can still be remarkably impacted by the thermal status of the upstream after dozens of years, which suggests that the

20 reconstruction of past climate changes on the glacier is crucial to better estimate the change of englacial thermal condition (e.g. Gilbert et al., 2014a).

### 6.3 Model limitations

Although our 2D, first-order, flowband model can account for part of the three-dimensional nature of LHG12 by parameterizing the lateral drag with glacier width variations, it cannot fully describe the ice flow along the $y$ direction, and is not able to account

for the confluence of glacier tributaries. The shape of the LHG12 glacier valley is described using a constant value for index $b$ (1.2; approximately "V" type cross-sections), which was determined from several traverse GPR profiles (Fig. 1). However, for real glaciers, the cross-sectional geometry profiles are generally complex, resulting in an inevitable bias when we idealize the glacier cross-sectional profiles by using power law functions across the entire LHG12. Although the regularized Coulomb friction law provides a physical relationship between the basal drag and sliding velocities, several parameters (e.g., $\lambda_{\max}$,

$m_{\max}$) still must be prescribed based on surface velocity observations. Another uncertainty could be from the spatially uniform geothermal heat flux that we assume in the model, as it may have a great spatial variation due to mountain topography (Lüthi and Funk, 2001). In addition, we can also improve our model ability by linking the water content in the temperate ice layer to a physical thermo-hydrological process in the future.

Due to the limitations of *in situ* shallow borehole ice temperature measurements, the surface thermal boundary condition in our temperature model is determined using a simple parameterization based on observations at three elevations (Fig. 1b). In addition, the parameterized surface thermal boundary condition only provides a rough estimate of the overall contributions of the heat from refreezing meltwater and ice flow advection. At this stage, we cannot simulate the actual physical process involved in the transport of near-surface heat from refreezing, which has been suggested by Gilbert et al. (2012) and Wilson and Flowers (2013). In our transient simulations, the glacier geometry and surface thermal parameters are assumed constant in time, which may lead to some unphysical model outputs. Due to a lack of long-term historical climate and geometry inputs, our transient simulations are only performed during 1961–2011 initialized with according diagnostic runs. However, the transient simulations in this study are mainly for seeking some possible reasons for the formation of the current temperature profile of the deep borehole. Therefore, we do not expect accurate model results from the transient experiments.

## 7   Conclusions

For the first time, we investigate the thermo-mechanical features of a typical valley glacier, Laohugou Glacier No.12 (LHG12), in Mt. Qilian Shan, which is an important fresh water source for the arid regions in western China. We assess the thermo-mechanical features of LHG12 using a two-dimensional thermo-mechanically coupled first-order flowband model based on available *in situ* measurements of glacier geometries, borehole ice temperatures, and surface meteorological and velocity observations.

Similar to other alpine land-terminating glaciers, the mean annual horizontal ice flow speeds of LHG12 are relatively low (less than 40 m a$^{-1}$). However, we observed large inter-annual variations in the ice surface velocity during the summer and winter seasons. Due to the release of heat from refreezing meltwater, the observed ice temperatures for the shallow ice borehole in the accumulation basin (site 3; Fig. 1) are higher than those at sites 1 and 2 at lower elevations, indicating the existence of meltwater refreezing. Thus, we parameterize the surface thermal boundary condition by accounting for the 20 m deep temperature instead of only the surface air temperatures. We observed that LHG12 has a polythermal structure with a temperate ice zone that is overlain by cold ice near the glacier base throughout a large region of the ablation area. Time-dependent simulations reveal that the englacial temperature becomes colder in recent two decades as a consequence of the shrink of accumulation area and rising surface air temperature.

Horizontal heat advection is important on LHG12 for bringing the relatively warm ice in the accumulation basin (due to the heat from refreezing meltwater) to the downstream ablation zone. In addition, vertical heat advection is important for transporting the near-surface cold ice downwards into the glacier interior, which "cools down" the ice. Furthermore, we argue that the strain heating of LHG12 also plays an important role in controlling the englacial thermal status, as suggested by Zhang et al. (2015). However, we also observed that simulated basal sliding (very small; $< 4$ m a$^{-1}$) contributes little to the thermal-mechanical configuration of LHG12.

The mean annual surface air temperature could serve as a good approximation for the temperatures of shallow ice, where seasonal climate variations are damped at cold and dry locations (Cuffey and Paterson, 2010, p. 404). However, for LHG12,

using the mean annual surface air temperature as the thermal boundary condition at the ice surface would predict an entirely cold glacier with very small ice flow speeds. For LHG12, a decline of ELA under a cold climate may assist an increase of the amount of refreezing latent heat in the accumulation basin, and therefore possibly raise the englacial temperature. Because warming is occurring on alpine glaciers in, for example, Mt. Himalayas and Qilian Shan, further studies of supra-glacial and near-surface heat transport are very important because they will affect the surface thermal conditions and, eventually, the dynamical behaviours of the glacier.

*Acknowledgements.* This work is supported by the National Basic Research Program (973) of China (2013CBA01801, 2013CBA01804), and the Technology Services Network Program (STS-HHS Program) of Cold and Arid Regions Environmental and Engineering Research Institute, Chinese Academy of Sciences (HHS-TSS-STS-1501). TZ is also supported by the National Natural Science Foundation of China (41601070). We are grateful to numerous people for their hard fieldwork. We thank the supports from the Qilian Shan Station of Glaciology and Ecologic Environment, Chinese Academy of Sciences (CAS). We thank Dr. Chen Rensheng and Dr. Liu Junfeng for providing the CAPD precipitation dataset. We thank Martin Lüthi, Andy Aschwanden and an anonymous reviewer for their thorough and constructive reviews which significantly improved the manuscript, as well as Oliver Gagliardini for his editorial work.

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

**Table 1.** Parameters used in this study

| Symbol | Description | Value | Unit |
| --- | --- | --- | --- |
| $\beta$ | Clausius-Clapeyron constant | $8.7 \times 10^{-4}$ | $\mathrm{K\,m^{-1}}$ |
| $g$ | Gravitational acceleration | 9.81 | $\mathrm{m\,s^{-2}}$ |
| $\rho$ | Ice density | 910 | $\mathrm{kg\,m^{-3}}$ |
| $\rho_{\mathrm{w}}$ | Water density | 1000 | $\mathrm{kg\,m^{-3}}$ |
| $n$ | Exponent in Glen's flow law | 3 | - |
| $\dot{\epsilon}_0$ | viscosity regularization | $10^{-30}$ | $\mathrm{a^{-1}}$ |
| $A_0$ | Flow law parameter | | |
| | when T $\leq$ 263.15 K | $3.985 \times 10^{-13}$ | $\mathrm{Pa^{-3}\,s^{-1}}$ |
| | when T $>$ 263.15 K | $1.916 \times 10^{3}$ | $\mathrm{Pa^{-3}\,s^{-1}}$ |
| $Q$ | Creep activation energy | | |
| | when T $\leq$ 263.15 K | 60 | $\mathrm{kJ\,mol^{-1}}$ |
| | when T $>$ 263.15 K | 139 | $\mathrm{kJ\,mol^{-1}}$ |
| $R$ | Universal gas constant | 8.31 | $\mathrm{J\,mol^{-1}\,K^{-1}}$ |
| $k$ | Thermal conductivity | 2.1 | $\mathrm{W\,m^{-1}\,K^{-1}}$ |
| $c_p$ | Heat capacity of ice | 2009 | $\mathrm{J\,kg^{-1}\,K^{-1}}$ |
| $L$ | Latent heat of fusion of ice | $3.35 \times 10^{-5}$ | $\mathrm{J\,kg^{-1}}$ |
| $T_0$ | Triple-point temperature of water | 273.16 | K |

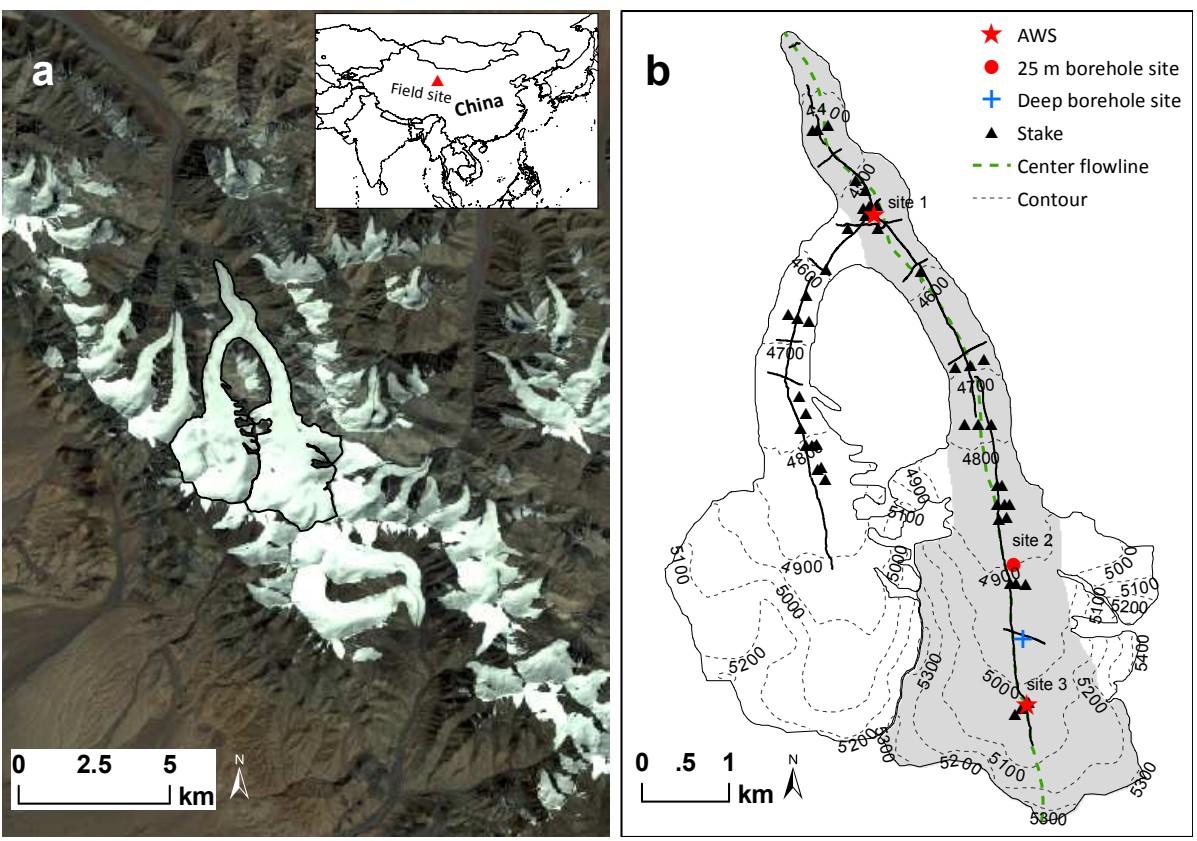

**Figure 1.** (a) The location of Laohugou Glacier No.12 (LHG12) in the west Mt. Qilian Shan, China. Underlain is a Landsat 5 TM image acquired on 22 September, 2011. (b) The solid and thick black lines indicate the ground-penetrating radar (GPR) survey lines. The shaded area denotes the main branch of LHG12, which neglects the west branch and all small tributaries. The green dashed line represents the center flowline (CL). Red stars indicate the locations of the automatic weather stations (AWSs) and the 25 m deep shallow boreholes (sites 1 and 3). A solid red circle represents the location of the shallow borehole at site 2, and a blue cross represents the location of the deep ice borehole. Black triangles show the positions of the stakes used for ice surface velocity measurements. The black contours are generated from SRTM DEM in 2000.

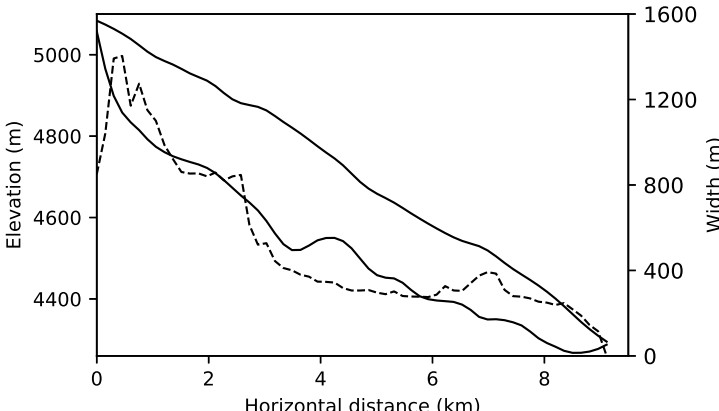

**Figure 2.** The glacier geometry of LHG12 along the CL. Solid lines show the glacier surface and bed elevations, while the dashed line shows the variation of glacier half widths along the CL.

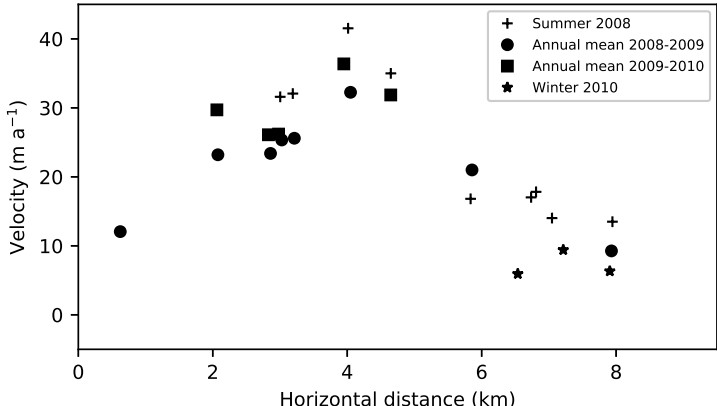

**Figure 3.** Measured ice surface velocities along the CL.

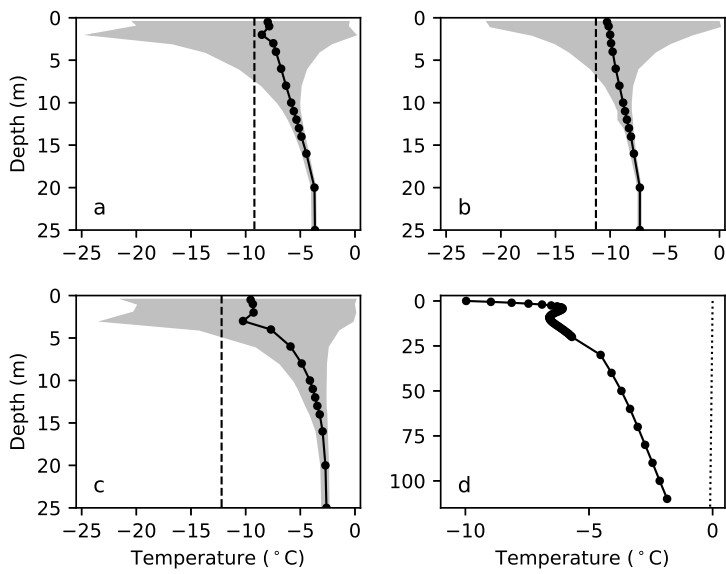

**Figure 4.** Ice temperature measurements from four ice boreholes. (a, b, c) Ice temperature measurements from the 25 m deep boreholes at site 1, 2, and 3, respectively. The black dots show the mean annual ice temperatures over the period of 2010 – 2011. The shaded areas show the yearly fluctuation range of the ice temperature. The dashed lines indicate the mean annual air temperature. (d) Measured ice temperatures in the deep borehole. The dotted line indicates the pressure-melting point (PMP) as a function of depth.

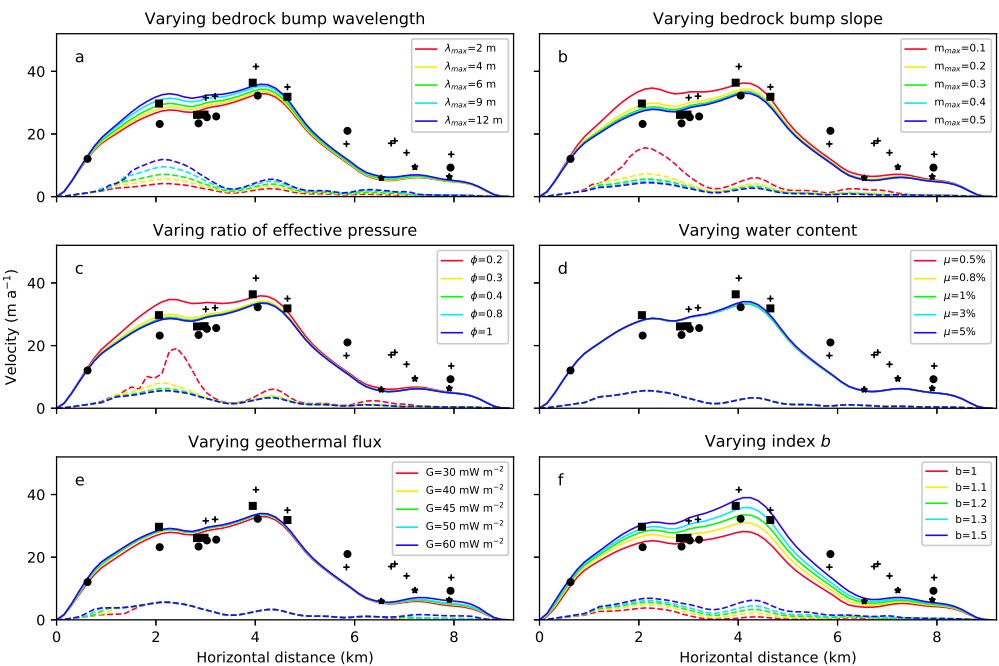

**Figure 5.** Sensitivity of modeled ice flow speeds to parameters along the CL with default parameters, i.e. $\lambda_{\max} = 4$ m, $m_{\max} = 0.3$, $\phi = 1$, $\mu = 0.03$, $G = 40$ mW m$^{-2}$, and $b = 1.2$. The solid and dashed lines indicate the modeled surface and basal sliding velocities, respectively. Markers indicate the measured ice surface velocities same as those shown in Fig. 3.

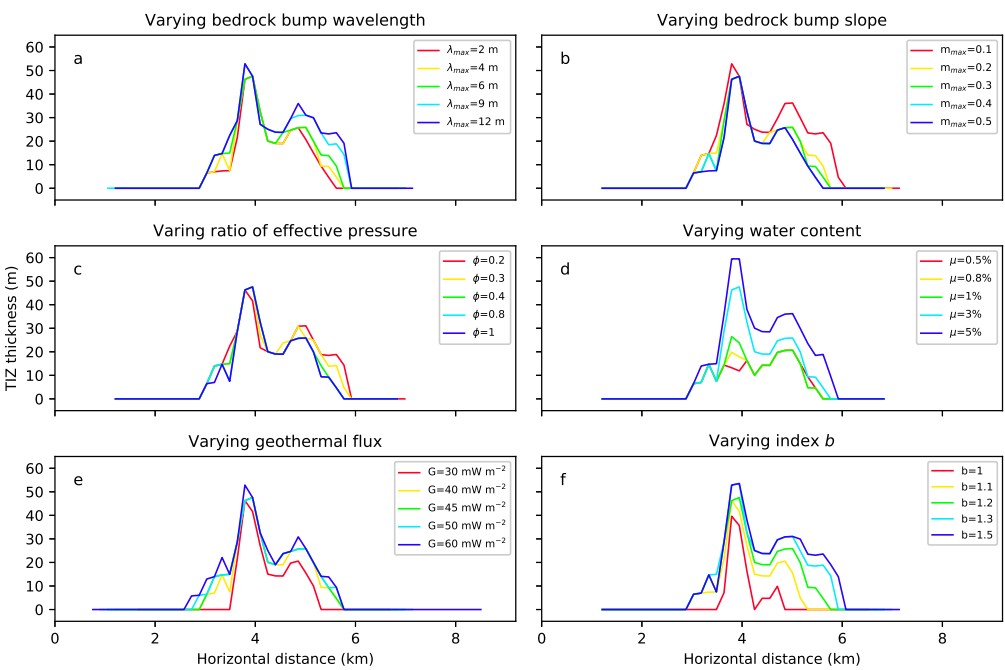

**Figure 6.** Sensitivity of modeled temperate ice thicknesses to parameters along the CL with default parameters, i.e. $\lambda_{\max} = 4$ m, $m_{\max} = 0.3$, $\phi = 1$, $\mu = 0.03$, $G = 40$ mW m$^{-2}$, and $b = 1.2$. The parameter settings are the same as described in Fig. 5.

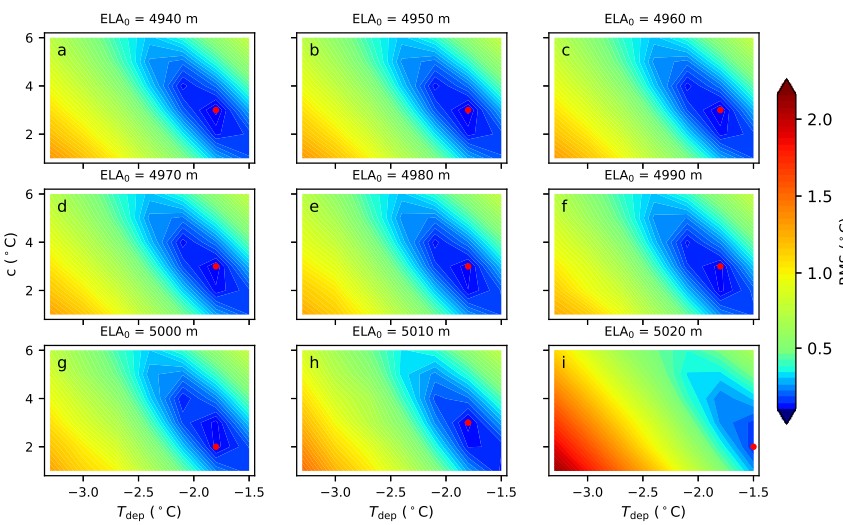

**Figure 7.** Root mean squares (RMS) of differences between measured and modeled temperature profiles in the deep borehole. The red circle indicates the minimum of RMS. The parameter $T_{\text{dep}}$ is varied from $-3.3\,^{\circ}\text{C}$ to $-1.5\,^{\circ}\text{C}$ with a step-size of $0.3\,^{\circ}\text{C}$, while $c$ is varied in the range of $1$–$6\,^{\circ}\text{C}$ with a step size of $1\,^{\circ}\text{C}$. The equilibrium line altitude (ELA) is fixed in each panel.

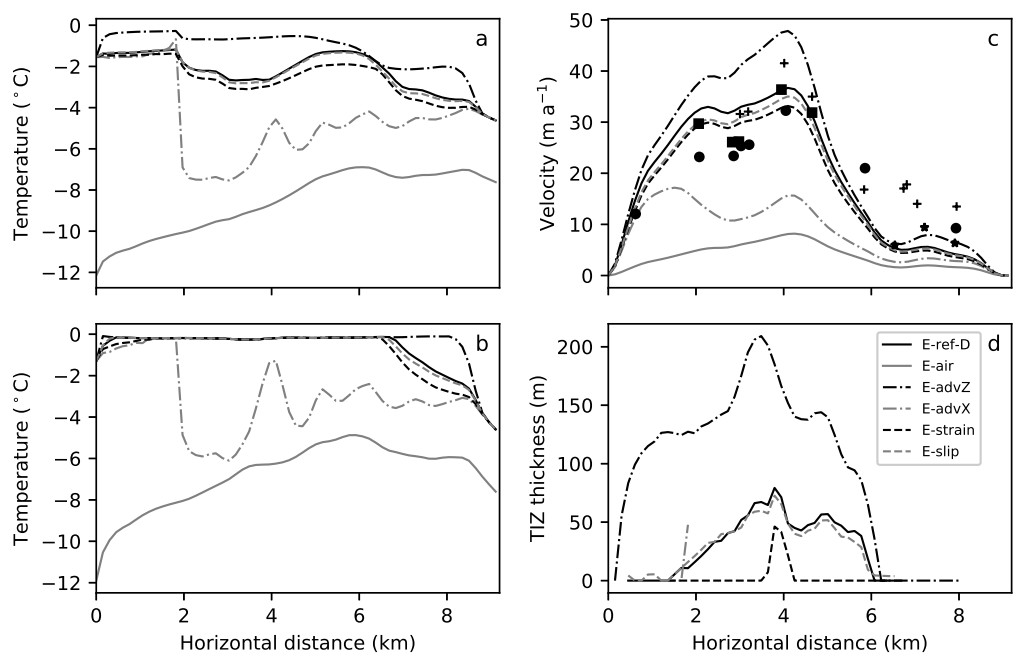

**Figure 8.** Modeled ice temperatures, velocities and TIZ thicknesses along the CL for diagnostic experiments E-ref-D (black solid line), E-air (gray solid line), E-advZ (black dash-dotted line), E-advX (gray dash-dotted line), E-strain (black dashed line), and E-slip (gray dashed line). (a) Modeled column mean ice temperatures. (b) Modeled basal ice temperatures. (c) Modeled surface horizontal velocities. (d) Modeled TIZ thickness.

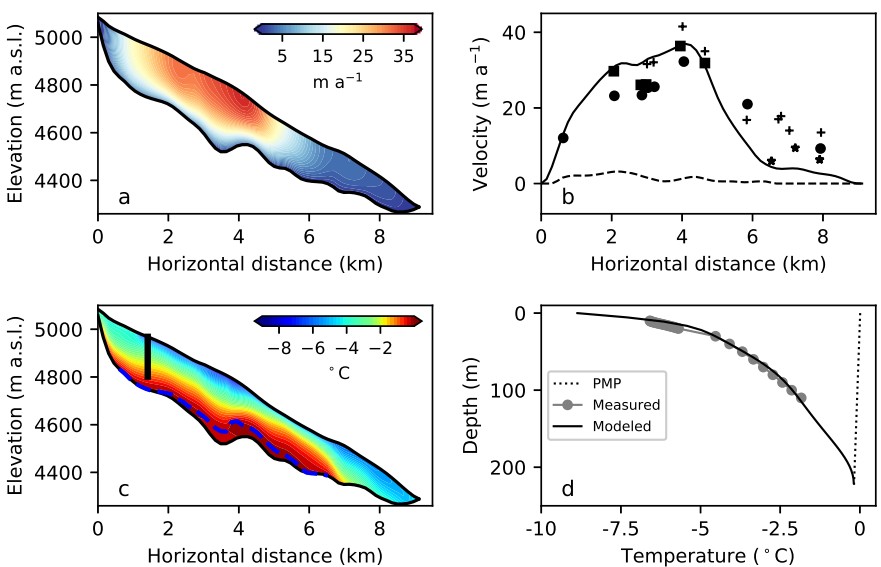

**Figure 9.** Comparison of measured and transiently modeled horizontal velocities and ice temperatures of LHG12. (a) Modeled distribution of horizontal ice velocity. (b) Measured (markers) and modeled surface (solid line) and basal (dashed line) horizontal velocities along the CL. (c) Modeled distribution of ice temperature. The blue dashed line indicates the CTS position, and the black bar indicates the location of the deep ice borehole. (d) Measured (dots) and modeled (solid line) ice temperature profiles in the deep borehole. Pressure-melting point is indicated by the dotted line.

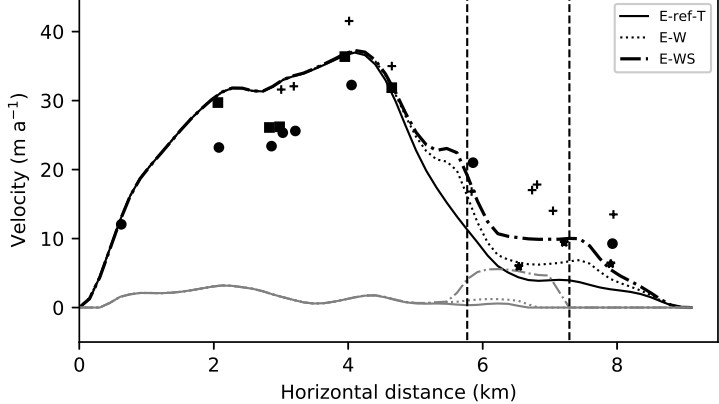

**Figure 10.** Modeled surface (black lines) and basal (gray lines) horizontal velocities along the CL for experiments E-ref-T (solid line), E-W (dotted line), and E-WS (dash-dotted line). The glacier widths in the zone of km 5.8 – 7.3 (bounded by the vertical dashed lines) are increased by 450 m for E-W and E-WS. In E-WS, we also include a basal sliding enhancement between km 5.8 – 7.3.

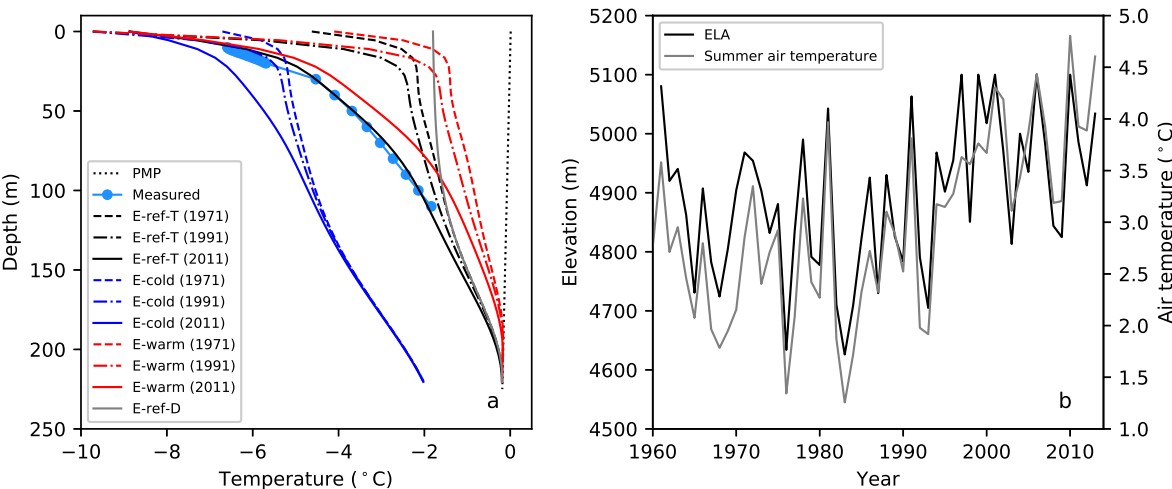

**Figure 11.** (a) Modeled temperature profiles in the deep borehole for experiments E-ref-T (black line), E-cold (blue line), E-warm (red line), and E-ref-D (gray line). Dots indicate the measured ice temperature profile. Pressure-melting point is indicated by the dotted line. (b) The variations of ELA and summer air temperature at 4200 m a.s.l. during 1961–2011.

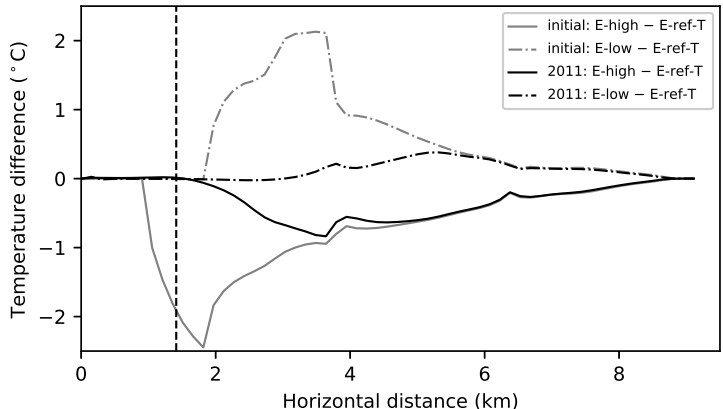

**Figure 12.** Differences of the column mean temperatures along the CL between the sensitivity experiments (E-high and E-low) and the reference experiment (E-ref-T). Gray lines indicate the initial temperature differences, whereas black lines indicate the temperature differences in 2011.