# Peer review of "An investigation of the thermo-mechanical features of Laohugou Glacier No.12 in Mt. Qilian Shan, western China, using a two-dimensional first-order flowband ice flow model"

_The Cryosphere, 2016_

## Short Comment (SC1) · 7 Mar 2016

This paper combines recent field measurements taken on a polythermal glacier on the Tibetan Plateau with results from a relatively sophisticated numeric model for ice temperature and flow. The authors vary the boundary conditions and the number of terms of the heat equation used in the model and compare the model output to annual ice velocities measured by stakes and englacial temperatures measured in a deep borehole. The work concludes that (a) strain heating and (b) refreezing meltwater in the percolation zone are the major controls on englacial temperatures and, consequently,

the velocity of this glacier.

In my opinion, the analysis and conclusions of the paper are sound and the ideas are presented clearly. Meltwater retention in firn is a topic of much current interest, especially on mountain glaciers where its effects can often influence basal temperatures. To my knowledge, thermal analyses of Himalayan glaciers are sparse, so this work would be a novel contribution for that alone.

The model appears to be state-of-the-art, and the limits of its application are discussed (omission of west branch). I think it would be beneficial to include a bit more discussion (and perhaps, but not necessarily, numeric estimation) of how inclusion of the secondary branch of the glacier would improve the results. (I think *why* it should improve the results is clear, but *how much* is not clear, and *where* is not in the expected locations.) To be clearer on my *where* point: it appears that the area of largest disagreement between measured/modeled velocities (5.3 to 7 km) occurs just upstream of the junction with the west branch (∼4500m elevation or ∼7 km). One would expect any effects of the west branch to be downstream of the junction.

This paper also appears to be the first presentation of the temperature data from the four boreholes, so a little more detail here would be appropriate. The description of the three shallow boreholes is more complete than for the deeper, and I would argue more important to the paper, borehole. For instance, how long were the sensors operational within the ice (were the temperatures able to equilibrate), and what is the error on the readings? How precise are the depths? The data in Figure 4(b) look smoother than I have usually seen from deep boreholes, leading to these questions.

My other suggestion is to improve the clarity of Figures 5, 6, and 7. Although the legends do indicate what is being plotted, they are small and encoded. This could be fixed easily by adding a title to each plot ("Varying bedrock bump wavelength") and/or adding this to the caption.

The caption for Figure 9(d) gets measured / modeled temperatures backwards.

I think this work is mature, interesting, and relevant, and I look forward to seeing it in published The Cryosphere.

---

## Referee Comment (RC1) · Anonymous Referee #1 · 31 Mar 2016

**Review of *"An investigation of the thermo-mechanical features of Laohugou Glacier No.12 in Mt. Qilian Shan, western China, using a two-dimensional first-order flow-band ice flow model"* by Wang et al.**

This paper analyses thermal and dynamical features of a glacier in Mt Qilian Shan (North Western China). The approach use a 2D thermo-mechanically coupled flow line model constrained on surface velocity and boreholes temperature measurements. Results show that the glacier is mostly cold with a potential basal temperate layer. The authors show that this layer is mostly due to strain heating and advection of warmer ice coming from the accumulation zone. The thermo-mechanical model is well constructed and use appropriate physics especially for calculating CTS position. The study brings insight in a much unknown region in glaciology.

However, this study clearly suffers of how boundary condition are addressed and of too much similarities with the already published *Zhang et al.* [2013]. Authors cannot performed a steady state simulation based on a punctual 20 m depth englacial temperature measurement, this will surely lead to a wrong temperature field. Indeed, near surface temperatures are probably not representative of the steady state regarding the recent context of atmospheric warming.

I don't think the current version of the manuscript deserves publication in The Cryosphere and needs at least a transient approach taking into account the glacier near-surface thermal evolution in response to climate variability before be resubmitted.

**General comments:**

- Although the graph are nicely prepared and the structure of the paper is clear, the too obvious similarities with *Zhang et al.* [2013] give the impression of reading exactly the same paper… The only change is the way that thermal boundary condition are addressed which is not a real improvement. I suggest to explore the transient state using available meteorological data to distinct this new study from *Zhang et al.* [2013].

- The thermal surface boundary condition should be better addressed. As I said above, the 20-meter-deep temperature is representative of the climatic forcing on the glacier energy balance during the previous year only. Using this temperature as boundary condition of a steady state simulation will lead to a temperature field probably far from the reality. The authors should, at least, try to develop a parametrization that linked $T_{sbc}$, $T_{air}$ and the ELA elevation based on the available observations on the glacier. I recommend to use in the ablation zone $T_{sbc} = T_{air} +$ constant and find the constant that

allows to match the measured $T_{20m}$ instead of using the approach of Wohlleben et al. [2009] which is very qualitative…

- I don't see any dependence of the sliding law to temperature. The authors seem to assume that sliding only depend of the effective pressure which is assumed to be uniformly proportional to the hydrostatic pressure in their study. This is very disputable, modeling sliding in cold area is very unusual in glaciology… Also, surface velocity measurement do not bring the evidence of sliding on this glacier. I think that removing sliding in the model still lead to modeled surface velocities under the measurements uncertainties (see next comment).

- Uncertainty on the surface velocity should be indicated to be able to discuss about the goodness of the fit and comparing velocity measurements at different periods. Is the difference between winter, summer and annual mean velocities are really significant?

- I note that the author have placed the ELA elevation to be able to "fit" their deep borehole data but is this ELA elevation really correspond to what is observed on the field?

**Specific comments**

I think you could write "englacial" instead of "en-glacial" everywhere.

*P 1 – line 1* : Remove first sentence

*P 1 – line 3* : Mt Qilian Shan located in

*P 1 – line 6* : match well (remove well before "but clearly")

*P1 – line 7 :* "because the flow branch is ignored" : this assertion is not really supported by anything in the paper and many other reason could be invoked

*P1 – line 7 :* "agree closely" : I don't agree, this is not a close match

*P1 – line 9 :*  highly : are highly

*P1 - line 9 :* Remove (for example … temperature)

*P1 – line 10 :* I don't think we can speak of the "work of Wohlleben et al. [2009]" talking about the qualitative assumption made is this paper

*P1 – line 13-14 :* Like (…) LHG12 : this is not true. Most important parameter are surface conditions including snow cover thickness and summer melting intensity.

*P1 – line 18-19 :* Sentence too long

*P2 – line 11-13 :* Bad example : what is the link with a full stokes model here ?

*P2 – line 13* : In addition = not appropriate here

*P2 - line 14* : "can be strongly influenced" : this is the main control !!

*P2 - line 22* :  remove "extremely"

*P 3 – line 12 :* explain why you are interested in parametrizing transverse profile ?

*P3 – line 18 -29 :* Give uncertainty on the measurement

*P4 – line 20-24* : There is no interest to detail the shape of the profile in the active layer

*P4 – line 28-29 :* Give the assumption of the model

*P5 – equation 6* : reference ?

*P6 – equation 10* : value of $\Gamma$ is not discussed

*P 8 - line 12 :* The authors claim a close match between model and observations at 80-90 m depth in the deep borehole: this is the point where the two curves (data and model) are just crossing! This not shows a good agreement between data and measurement.

*P 8 – line 33* : Is there moulin on this cold glacier ?

*P11 – line 1-2* : Remove sentence

---

## Referee Comment (RC2) · M.P. Lüthi (Referee) · 30 Apr 2016

**Review: Wang et al, tc-2016-38**

Dear colleagues,

This is a very nice and comprehensive study of a remote glacier in a barely investigated mountain range. I find it especially valuable since a very interesting set of field data are presented which are interpreted with the help of a numerical model. The modeling study is comprehensive, and the investigation of the importance of the individual advective terms, dissipation and basal motion (Section 4.4) is insightful.

My recommendation is to publish the manuscript after the comments below have been taken into account.

Sincerely, Martin Lüthi

**Specific comments**

Leave away colons (:) before equations, this is not usual in The Cryosphere.

You should decide on one version of English. Now there are "modeled" and "modelled" in the same sentence.

p1,1   "see" could be omitted

p3,11   also give the slope angle in degrees, i.e. $4.6°$.

p3,12   $L$ is often used for the glacier length, $y$ would be more common for a transverse coordinate.

p3,19   also indicate distance from terminus, or along-profile.

p5,13   omit ":", maybe writing "following Flowers et al. (2011)

p5,16   "horizontal diffusion is parametrized by glacier width" is quite opaque. Please explain what you are doing, since this is not standard. This seems to be middle term in the parentheses, but it is not clear where this comes from. Does this somehow parametrize lateral diffusion (along the $y$-Axis)? But then, why would the longitudinal velocity gradient $dT/dx$ play a role? Please explain this in detail (maybe in an appendix).
Overall, it seems advantageous to ignore heat flow in $y$-direction (i.e. leave away the problematic term in Equation (6), since nothing is known about the boundary conditions there.

p5,26   What happens with water produced by dissipation? Does this stay in the ice, or does it drain at a certain volume ratio? Is a balance equation for the water content, or the enthalpy, solved?

p6,3   Even if the model is described elsewhere in detail, the main characteristics should be given here: solution method (finite difference, finite element, ...), discretization (element type, mesh size), solution method (solver, time-stepping, CFL condition) etc., and maybe some implementation details (solver libraries used, maybe Matlab, etc...).

p6,7   Parentheses should be adapted using \left( and \right)

p6,11   Strictly, this should be $\sigma_n - p_w$ using the normal stress on the bed, which might be quite different from the overburden calculated with the local vertical ice thickness. In which direction is $H$ measured, vertically (along $z$), or perpendicular to the ice surface?

p6,30 This boundary condition is valid for cold ice, but what is used in temperate ice? There, any geothermal heat will contribute to melting.

p7,2 I assume that the $G$-term is not very important for the model results. In mountain topography, the geothermal heat flux can vary a lot on short spatial scales, so the importance of this should be at least discussed.

p7,22 So, the water content is assumed constant throughout the temperate ice? This is problematic and will obviously introduce some inaccuracies.

p8,3 "compare to"

p8,6 The omission of convergent flow is only one possible (and likely) explanation, but there might be others, e.g. basal motion. This statement should be made more carefully.

p8,7 Here you should qualify "the modeled basal sliding velocities", IIUC. The reality, again, could be that basal sliding is much higher there. This could be elaborated upon in the Discussion.

p8,8 "observed": this is confusing, as you talk about model results. Better say: "the model predicts"

p8,9 add space between "110m"

p8,10 "ice fluxes" (not "ice flows")

p8,13 More important than matching temperatures would be a discussion of the heat fluxes. While the measurements show constant fluxes below $50\,\mathrm{m}$ depth below the surface, the model shows zones of warming and cooling (bends in the temperature profile). It would be important to understand the reason for these excursions from a straight line, is the shape of this profile due to advection, dissipation, or due the temperature history?

Closer to the surface (above $50\,\mathrm{m}$ depth) the measured gradient is much higher, which might reflect the thermal properties of the firn in a steady state (lower conductivity $k$). Since ice conductivity is assumed everywhere in the model, this might explain the difference there (cf. Fig. 5 in Lüthi and Funk (2001) for a theoretical temperature profile with firn).

p8,27 It would be helpful to also show a graph of TIZ thickness (a second panel in Fig. 8c). It appears that the bed is temperate almost everywhere in the blue and green model runs, but with very small TIZ.

p8,31 "above" (leave away "in")

p9,11 ff instead of "drop" and "remove" you could consistently use "neglect" or "leave away"

p9,25 qualify "basal sliding" by "modeled"

p9,28 leave away "higher-order"

p10,2 "physically" should be "physical"

p10,9 consolidate the two citations

p10,11 Past changes can have a very important impact (see for example Lüthi et al. (2015)), as are warming processes in the firn (e.g. Machguth et al. (2016))

p11,16 Replace "e.g." with "of" (these are not just examples, but an exhaustive list of measurements used in the study).

p11,21 No need to show the symbol "(u)" here again (leave away).

Fig 1 A nice overview photograph would help setting the scene for this remote glacier that most readers won't know.

Fig 2 same labels on the horizontal axis of Figs. 3 and 4 would ease of comparison.

Fig 8d Caption: modeled and measured (lines vs symbols) should be interchanged.

**References**

Lüthi, M. P. and Funk, M. (2001). Modelling heat flow in a cold, high altitude glacier: interpretation of measurements from Colle Gnifetti, Swiss Alps. *Journal of Glaciology*, 47(157):314–324.

Lüthi, M. P., Ryser, C., Andrews, L. C., Catania, G. A., Funk, M., Hawley, R. L., Hoffman, M. J., and Neumann, T. A. (2015). Heat sources within the Greenland Ice Sheet: dissipation, temperate paleo-firn and cryo-hydrologic warming. *The Cryosphere*, 9:245–253.

Machguth, H., MacFerrin, M., van As, D., Box, J. E., Charalampidis, C., Colgan, W., Fausto, R. S., Meijer, H. A. J., Mosley-Thompson, E., and van de Wal, R. S. W. (2016). Greenland meltwater storage in firn limited by near-surface ice formation. *Nature Climate Change*.

---

## Author Comment (AC1) · 8 Jun 2016

article

amsmath amssymb graphicx a4wide

[Figure]

**Response to K. Poinar**

Yuzhe Wang

We would like to thank Dr. K. Poinar for giving constructive and encouraging comments on our paper; they were very helpful to improve our manuscript. Our responses to all the comments are given below. The original reviewer's comments are given in italic, and our responses are given directly below in regular.

**Specific comments**

*The model appears to be state-of-the-art, and the limits of its application are discussed (omission of west branch). I think it would be beneficial to include a bit more discussion (and perhaps, but not necessarily, numeric estimation) of how inclusion of the secondary branch of the glacier would improve the results. (I think \*why\* it should improve the results is clear, but \*how much\* is not clear, and \*where\* is not in the expected locations.) To be clearer on my \*where\* point: it appears that the area of largest disagreement between measured/modeled velocities (5.3 to 7 km) occurs just upstream of the junction with the west branch (âĹij4500m elevation or âĹij7 km). One would expect any effects of the west branch to be downstream of the junction.*

Thanks for your good suggestion. The underestimation may possibly result

[Figure]

**Fig. 1.** Modeled ice velocities for experiments E-ref (blue line), E-W (red line), and E-WS (green line). The glacier widths in the zone bounded by the vertical dashed lines are uniformly increased by 450 m.

from the neglect of the convergent flow from the west branch and an enhanced basal sliding which is not captured by our model in the confluence area (see our responses to the other two reviewers). To verify this hypothesis, we conduct two other experiments, E-W and E-WS. In E-W the glacier widths are increased by 450 m at km 5.8 – 7.3 as a proxy of including the impact of the convergent flow from the west branch (Fig. 1). In E-WS, except for the same glacier width increase as in E-W, we also increase $\lambda_{max}$ by 200% and decrease $m_{max}$ by 60% for accelerating the basal sliding at km 5.8 – 7.3 (Fig. 1). We can clearly find that while both factors have a non-negligible contribution to the model results, the basal sliding may play a bit more important role in the confluence area. The basal sliding velocities in experiment E-WS can be raised to 9.5 m a$^{-1}$ at km 6.2. The mean ice surface velocities modeled by E-W and E-WS in the distance of km 5.3 – 9.1, are larger than those of E-ref by 2.1 m a$^{-1}$ and 4.9 m a$^{-1}$, respectively. This indicates a need of considering glacier flow branches and spatially variable sliding law parameters in real glacier modeling studies.

*This paper also appears to be the first presentation of the temperature data from the four boreholes, so a little more detail here would be appropriate.*

*The description of the three shallow boreholes is more complete than for the deeper, and I would argue more important to the paper, borehole. For instance, how long were the sensors operational within the ice (were the temperatures able to equilibrate), and what is the error on the readings? How precise are the depths? The data in Figure 4(b) look smoother than I have usually seen from deep boreholes, leading to these questions.*

Good suggestions. We now have added more details about the measurement of deep borehole temperatures.

"To determine the englacial thermal conditions of LHG12, we drilled a deep ice core (167 m) in the upper ablation area of LHG12 (approximately 4971 m a.s.l., Fig. 1). In October 2011, ice temperature were measured to a depth of approximately 110 m using a thermistor string after 20 days of the drilling, as shown in Fig. 4d. The string consists of 50 temperature sensors with a vertical spacing of 0.5 m and 10 m at the ice depths of $0-20$ m and $20-110$ m, respectively. The accuracy of the temperature sensor is around $\pm 0.05\,°C$ (Liu et al., 2009)."

*My other suggestion is to improve the clarity of Figures 5, 6, and 7. Although the legends do indicate what is being plotted, they are small and encoded. This could be fixed easily by adding a title to each plot ("Varying bedrock bump wavelength") and/or adding this to the caption.*

This is a good idea, and we now have added the title for each panel.

*The caption for Figure 9(d) gets measured / modeled temperatures backwards.*

Fixed.

---

## Author Comment (AC2) · 8 Jun 2016

The comment was uploaded in the form of a supplement:
http://www.the-cryosphere-discuss.net/tc-2016-38/tc-2016-38-AC2-supplement.pdf

---

## Author Comment (AC3) · 8 Jun 2016

**Response to Anonymous Referee #1**

Yuzhe Wang

We would like to thank the anonymous referee #1 for giving constructive comments on our paper. We have responded each comment with great care. The original comments of the reviewer are given in italic, and our responses are given directly below in regular.

**General comments**

*Although the graph are nicely prepared and the structure of the paper is clear, the too obvious similarities with Zhang et al. [2013] give the impression of reading exactly the same paper The only change is the way that thermal boundary condition are addressed which is not a real improvement. I suggest to explore the transient state using available meteorological data to distinct this new study from Zhang et al. [2013].*

It's true both studies share a few of similarities. After the attempt of Zhang et al. (2013) on the East Rongbuk Glacier, Mt. Qomolangma (Everest), we've been curious about the thermo-mechanical features of other typical Tibetan mountain glaciers. Does the climate warming really have a great impact on these glaciers and how much are these impacts? The East Rongbuk Glacier is at the southern edge of Tibetan Plateau. The one we get interested this time, Laohugou Glacier No. 12, is, however, at the northeastern edge of Tibetan Plateau. Despite the big different locations and climate backgrounds, both glaciers have been taken as fully cold for quite a long time by our China glaciological community. We hope that, by using similar numerical techniques, we could possibly get some interesting findings that can guide us to a big picture of Tibetan glacier changes. For example, does this 2D flowband model really work for mountain valley glaciers (we can save a lot of field efforts and money if it or something similar works)? If yes, how much can we rely on it? if not, how can we improve it? But first we should test it at different locations. That's the main reason we use a similar model approach and study method to Zhang et al. (2013).

We agree that the past climate change may have a great influence on the glacier velocities and temperature field. The difference between our diagnostic model results and the observations can be either from the assumptions of the model physics or the transient state of glacier change. We really wish we could do the transient study for LHG12 (and the East Rongbuk Glacier). Despite some previous expeditions in 1970s and 1980s, there is very few long-term series of meteorological data available in this area. The glaciological station was established in 2008 and we do not have the radar gemotry data of 2008 either. Thus, our aim is to investigate the current

thermo-mechanical state by neglecting the transient impacts. We know by doing this there will be some uncertainties in our model results. We assume the transient effect in past years is stable and our thermal steady-state assumption is effective. We believe that our thermal steady-state model can capture some characteristics of glacier behaviours within the range of historical changes, and that our conclusion that LHG12 is now polythermal should be robust. To be as cautious as we can, we avoid showing precise number of, like, temperate ice zone lengths and thickness in both the abstract and the conclusions.

*The thermal surface boundary condition should be better addressed. As I said above, the 20-meter-deep temperature is representative of the climatic forcing on the glacier energy balance during the previous year only. Using this temperature as boundary condition of a steady state simulation will lead to a temperature field probably far from the reality. The authors should, at least, try to develop a parametrization that linked $T_{sbc}$, $T_{air}$ and the ELA elevation based on the available observations on the glacier. I recommend to use in the ablation zone $T_{sbc} = T_{air} + constant$ and find the constant that allows to match the measured $T_{20m}$ instead of using the approach of Wohlleben et al. [2009] which is very qualitative*

As suggested by the reviewer, we now prescribe the $T_{sbc}$ in the ablation area by a simple parameterization $T_{sbc} = T_{air} + c$, where $c$ is a tuning parameter including the impacts of both the surface energy budget and the steady-state temperature Gilbert et al. (2010). We vary the values of $c$ from 0 to 6 K (with a step-size of 0.2 K) and compare the modeled 20 m borehole temperatures with in-situ annual measurements at site 1 and 2 (Fig. 1). As shown in Fig. 1b (in manuscript), site 1 is located at the center of the confluence area where the convergent flow from the west branch joins the mainstream. Thus, at site 1 it is difficult to find a good $c$ value that predicts close temperature comparisons to the observations. We therefore determine the $c$ value (1.6 K) based on the fittings between the modeled and observed ice temperature data at site 2.

[Figure]

Figure 1: Sensicitity experiments of the tuning parameter $c$ by comparing the measured (black dotted lines) and modeled (coloured lines) 20 m borehole temperatures at sites 1 (a) and 2 (b). The step-size of varying the $c$ value is 0.2 K.

In addition, we also compare the differences between the new (E-new, (Gilbert et al., 2010)) and the old (E-old, (Wohlleben et al., 2009)) parameterizations of the thermal surface boundary conditions (Fig. 2). It shows that the two experiments produce very similar results in terms of modeled ice surface velocities, basal sliding velocities, temperate ice zones, and temperature profiles at the deep borehole (Fig. 2b, c, d). As can be expected, the modeled column mean and basal temperatures in the distance of km 5.0 – 9.1 demonstrate large differences due to the different parameterizaitions in the ablation area (Fig. 2a).

[Figure]

Figure 2: Modeled ice temperatures and velocities for experiments E-old (blue line) and E-new (red line). (a) Modeled column mean (solid lines) and basal (dashed lines) ice temperatures along the center flowline. (b) Modeled surface (solid lines) and basal (dashed lines) ice velocities along the CL. The symbols show the measured ice surface velocities same as in the manuscript. (c) Modeled CTS position (solid lines) and TIZ thickness (dashed lines). The black bar shows the location of the deep ice borehole. (d) Measured (dots) and modeled (coloured lines) ice temperature profiles for the deep borehole. The dotted line shows the pressure-melting point as a function of ice depth.

*I don't see any dependence of the sliding law to temperature. The authors seem to assume that sliding only depend of the effective pressure which is assumed to be uniformly proportional to the hydrostatic pressure in their study. This is very disputable, modeling sliding in cold area is very unusual in glaciology Also, surface velocity measurement do not bring the evidence of sliding on this glacier. I think that removing sliding in the model still lead to modeled surface velocities under the measurements uncertainties (see next comment).*

It's not true. The sliding events are certainly a result of the existence of temperate ice. At the ice-bed interface, we prescribe a non-slip boundary condition where ice is frozen to the bed (cold ice) and a Coulomb friction law where ice is temperate, i.e., the ice temperature reaches the local pressure-melting point. We have clarified this in p6–line9.

*Uncertainty on the surface velocity should be indicated to be able to discuss about the goodness of the fit and comparing velocity measurements at different periods. Is the difference between winter, summer and annual mean velocities are really significant?*

We agree the reviewer that the uncertainties of the ice velocity data are important for evaluating our model results. We estimate the data uncertainty below 1 m a$^{-1}$. But the stakes are not exactly located on the center flowline, which may also bring some unknown uncertainties. The summer (2008) and winter (2010) velocity data we have are not from a single year. They cannot be exactly compared. But from the only overlapped point we have (Fig. 3 in the manuscript), the difference between winter and summer values is non-negligible – it could be up to around 50%. We have added the uncertainties of GPS positioning and the calculated velocities in p3 – line23-24.

*I note that the author have placed the ELA elevation to be able to "fit" their deep borehole data but is this ELA elevation really correspond to what is observed on the field?*

The ELA was identified from the Landsat image on September 6, 2011, which is quite close the time (October 1–6, 2011) we drilled the borehole.

**Specific comments**

*I think you could write "englacial" instead of "en-glacial" everywhere.*

Changed.

*P1 line 1: Remove first sentence*

Removed.

*P1 line 3: Mt Qilian Shan located in*

Changed.

*P1 line 6: match well (remove well before "but clearly")*

Changed.

*P1 line 7: "because the flow branch is ignored": this assertion is not really supported by anything in the paper and many other reason could be invoked*

It's correct that the neglect of the flow branch may be one of many reasons. We have conducted two other experiments by increasing the glacier width as a proxy of convergent effects of the west branch and by adjusting the friction sliding parameters at the confluence area. We found that both basal sliding and convergent effect can largely influence the ice surface velocities in that area. We now add several

sentences in p9–line3-10 and also include an additional figure (Fig. 9 in manuscipt)

*P1 line 7: "agree closely" : I don't agree, this is not a close match*

From our point of view, it's quite close, given the facts of the sparse observations and the simplified 2D model we use. But as the reviewer suggested, we now remove "closely".

*P1 line 9: were highly: are highly*

Corrected.

*P1 - line 9: Remove (for example temperature)*

Removed.

*P1 line 10: I don't think we can speak of the "work of Wohlleben et al. [2009]" talking about the qualitative assumption made is this paper*

*P1 line 13-14: Like (...) LHG12: this is not true. Most important parameter are surface conditions including snow cover thickness and summer melting intensity.*

We thank the reviewer for this comment. We now change the sentence as "strain heating is an important parameter controlling the englacial thermal structure in LHG12." .

*P1 line 18-19: Sentence too long*

Changed. Now the sentence becomes
"Located on the northeastern edge of the Tibetan Plateau (36 – 39 °N, 94 – 104 °E), Mt. Qilian Shan (MQS) develops 2051 glaciers covering an area of approximately 1057 km$^2$ with a total ice volume of approximately 50.5 km$^3$ (Guo et al., 2014, 2015). Meltwater from MQS glaciers is a very important water resource for the agricultural irrigation and socio-economic development of the oasis cities in northwestern China."

*P2 - line 11-13: Bad example: what is the link with a full stokes model here?*

This example was mainly for underlining the importance of temperate ice. But we agree with the reviewer. The sentence is now removed. Lines 10–13 have been changed to:
"The temperature distribution of a glacier primarily controls the ice flow rheology, englacial hydrology, and basal sliding conditions (Blatter and Hutter, 1991; Irvine-Fynn et al., 2011; Schäfer et al., 2014). A good understanding of the glacier thermal regime is important for predicting glacier response to climate change (Wilson et al., 2013; Gilbert et al., 2015), improving glacier hazard analysis (Gilbert et al., 2014a), and reconstructing past climate histories (Lüthi and Funk, 2001; Gilbert et al., 2010)."

*P2 line 13: In addition = not appropriate here*

This line has been changed as shown in above.

*P2 - line 14: "can be strongly influenced": this is the main control!!*

We have corrected it and add some corresponding references. Now it reads:

"The thermal regime of a glacier is mainly controlled by the surface thermal boundary conditions (e.g., Gilbert et al., 2014b; Meierbachtol et al., 2015). For example, near-surface warming from refreezing melt-water and cooling from the cold air of crevasses influence the thermal regimes of glaciers (Wilson and Flowers, 2013; Wilson et al., 2013; Gilbert et al., 2014a)."

*P2 - line 22: remove "extremely"*

We consider the LHG12 as an extremely continental-type glacier according to the classification of Shi and Liu (2000) who categorized the China glaciers into three types: the maritime (temperate) type, sub-continental (sub-polar) type and extremely continental (polar) type. We prefer to keep "extremely" as an identifier to the sub-continental type.

*P3 line 12: explain why you are interested in parametrizing transverse profile?*

The LHG12 is a valley glacier which is confined to channels with lateral drag exerted by the valley walls. As you all know, the lateral drag has a remarkable impact on glacier dynamics. To account for the lateral drag in a 2D ice flow model, we may either use a so-called "shape factor" proposed initially by Nye (1965) and impressed again recently by Adhikari and Marshall (2012) or make a parameterization based on glacier widths at all depths (Pimentel et al., 2010; Zhang et al., 2013). By parametering the transverse profile based on GPR measurements, we can derive the widths of glacier cross-sections and parameterize lateral drags at different depths (section 3.1). We now add an additional sentence for explanation in p3–line11-13.

*P3 line 18 -29: Give uncertainty on the measurement*

We have added a description of the uncertainties on the measurements.

"We measured the stake positions using a real-time kinematic (RTK) fixed solution by a South Lingrui S82 GPS system (Liu et al., 2011). The accuracy of the GPS positioning is an order of a few centimeters and the uncertainty of the calculated ice surface velocities is estimated to be less than 1 m a$^{-1}$."

*P4 line 20-24: There is no interest to detail the shape of the profile in the active layer*

We have deleted the description of temperature variations in the active layer.

*P4 line 28-29: Give the assumption of the model*

We now add the assumptions. "By assuming the vertical normal stress as hydrostatic and neglecting the bridging effects (Pattyn, 2002), the equation for momentum balance is given as".

*P5 equation 6: reference?*

It's (Pattyn, 2002). Now added.

*P6 equation 10: value of $\Gamma$ is not discussed*

$\Gamma = 0.84 m_{\max}$. Now added.

*P8 - line 12: The authors claim a close match between model and observations at 80-90 m depth in the deep borehole: this is the point where the two curves (data and model) are just crossing! This not shows a good agreement between data and measurement.*

LHG12 is a very large valley glacier. Though a lot of field work have been taken on this glacier, the *in-situ* observations are still sparse and temporally discontinuous. This is also one of many reasons that we didn't try 3D Stokes ice flow model. It's true that there are still some obvious disagreements between modeling results and *in-situ* observations. But given these poorly datasets, we are actually quite happy about the curves. However, as the reviewer suggested, we have removed the word "close".

*P8 line 33: Is there moulin on this cold glacier?*

Yes, we observed several moulins in the middle ablation area in 2009 and 2014.

*P11 line 1-2: Remove sentence*

Removed.

**References**

Adhikari, S. and Marshall, S. J.: Parameterization of lateral drag in flowline models of glacier dynamics, Journal of Glaciology, 58, 1119–1132, 2012.

Gilbert, A., Wagnon, P., Vincent, C., Ginot, P., and Funk, M.: Atmospheric warming at a high-elevation tropical site revealed by englacial temperatures at Illimani, Bolivia (6340 m above sea level, 16°S, 67°W), Journal of Geophysical Research, 115, D10 109, 2010.

Nye, J.: The flow of a glacier in a channel of rectangular, elliptic or parabolic cross-section, Journal of Glaciology, 5, 661–690, 1965.

Pattyn, F.: Transient glacier response with a higher-order numerical ice-flow model, Journal of Glaciology, 48, 467–477, 2002.

Pimentel, S., Flowers, G. E., and Schoof, C. G.: A hydrologically coupled higher-order flow-band model of ice dynamics with a Coulomb friction sliding law, Journal of Geophysical Research: Earth Surface, 115, 2010.

Shi, Y. and Liu, S.: Estimation on the response of glaciers in China to the global warming in the 21st century, Chinese Science Bulletin, 45, 668–672, 2000.

Wohlleben, T., Sharp, M., and Bush, A.: Factors influencing the basal temperatures of a High Arctic polythermal glacier, Annals of Glaciology, 50, 9–16, 2009.

Zhang, T., Xiao, C., Colgan, W., Qin, X., Du, W., Sun, W., Liu, Y., and Ding, M.: Observed and modelled ice temperature and velocity along the main flowline of East Rongbuk Glacier, Qomolangma (Mount Everest), Himalaya, Journal of Glaciology, 59, 438–448, 2013.

---

## Author Comment (AC4) · 8 Jun 2016

**Response to Referee #2**

Yuzhe Wang

We would like to thank Martin Lüthi for giving insightful and constructive comments on our paper; they were very helpful to improve our manuscript. Our responsees to all the comments are given below. The original comments of the reviewer are given in italic, and our responses are given directly below in regular.

**Specific comments**

*Leave away colons (:) before equations, this is not usual in The Cryosphere.*

We have deleted colons before equations and have reformulated the sentences if necessary.

*You should decide on one version of English. Now there are "modeled" and "modelled" in the same sentence.*

We now use "modeled" and "modeling" in the manuscript.

*p1,1 "see" could be omitted*

Changed. We have also removed "see" appeared in other similar case.

*p3,11 also give the slope angle in degrees, i.e. 4.6°.*

Changed.

*p3,12 L is often used for the glacier length, y would be more common for a transverse coordinate.*

We have changed the equation to $z = aW(z)^b$.

*p3,19 also indicate distance from terminus, or along-profile*

We have added the distance information. The sentence now reads:

"All stakes were located in the distance between km 0.6 – 7.9 along the CL (Fig. 3), spanning an elevation range of 4355 – 4990 m.a.s.l. (Fig. 1)."

*p5,13 omit ":", maybe writing "following Flowers et al. (2011)"*

The sentence has been reformulated to "we parameterize the lateral drag, $\sigma'_{xy}$, as a function of the flow-band half width, $W$, following Flowers et al. (2011)".

*p5,16 "horizontal diffusion is parametrized by glacier width" is quite opaque. Please explain what you are doing, since this is not standard. This seems to be middle term in the parentheses, but it is not clear where this comes from. Does this somehow parametrize lateral diffusion (along the y-Axis)? But then, why would the longitudinal velocity gradient dT/dx play a role? Please explain this in detail (maybe in an appendix).*

*Overall, it seems advantageous to ignore heat flow in y-direction (i.e. leave away the problematic term in Equation (6), since nothing is known about the boundary conditions there.*

We directly use the parameterization of heat diffusion in $y$ from Pattyn (2002) (Equation (16)) therein). It's just a rough assumption. We didn't check with F. Pattyn for the details of the mathematical derivation. The thoughts behind it, by our understanding, are from (1) assuming $\partial T/\partial x$ has a linear relationship with $\partial T/\partial y$, $\partial T/\partial y = \partial W/\partial x \times \partial T/\partial x$; (2) assuming $\partial^2 T/\partial y^2 = 1/W \times \partial T/\partial y$.

As suggested by the reviewer, we now have removed the diffusion term in $y$. As shown below, this diffusion along $y$ (E-yDiffu) has very limited impact on the model results, compared with the case without it (E-ref). Thus, we can indeed ignore this term in the 2D ice temperature model.

[Figure]

Figure 1: Modeled ice temperatures and velocities for experiments E-ref (blue line, without diffusion in y-axis) and E-yDiffu (red line, with parameterized diffusion in y-axis)

*p5,26 What happens with water produced by dissipation? Does this stay in the ice, or does it drain at a certain volume ratio? Is a balance equation for the water content, or the enthalpy, solved?*

We thank the reviewer for these very good questions! Sorry to admit that we've not considered those problems so far yet. We assume a constant water content in the temperate ice layer. But we haven't yet included a thermo-hydrological model. An enthalpy scheme for the polythermal glacier with a balance equation is under development.

*p6,3 Even if the model is described elsewhere in detail, the main characteristics should be given here: solution method (finite difference, finite element, …), discretization (element type, mesh size), solution method (solver, time-stepping, CFL condition) etc., and maybe some implementation details (solver libraries used, maybe Matlab, etc…).*

The numerical implementations are the same as described in Zhang et al. (2013). We now have added a sub-section introducing the main features of the numerical solutions in p7–line19-25.

*p6,7 Parentheses should be adapted using \left( and \right)*

Corrected.

*p6,11 Strictly, this should be $\sigma_n - P_w$ using the normal stress on the bed, which might be quite different from the overburden calculated with the local vertical ice thickness. In which direction is H measured, vertically (along z), or perpendicular to the ice surface?*

Here $H$ is vertical to the ice-bed interface (along $z$). The effective pressure used in the friction law is defined as the ice overburden pressure (see Gagliardini et al. (2007)), not the normal stress.

*p6,30 This boundary condition is valid for cold ice, but what is used in temperate ice? There, any geothermal heat will contribute to melting.*

If there is a temperate layer at the glacier base, two cases must be distinguished. For the melting case where cold ice flows into the temperate ice, we assume a negligible water content, and the ice temperature gradient at the CTS equals to the Clausius-Clapeyron gradient ($\beta$). For the freezing case where temperate ice flows into the cold ice, the latent heat released due to refreezing must be taken into account. We assume an ice temperature gradient at the CTS following Funk et al. (1994):

$$\frac{\partial T}{\partial z} = -\frac{Q_r}{k} + \beta. \tag{1}$$

The above description has been illustrated in the section 3.2.

*p7,2 I assume that the G-term is not very important for the model results. In mountain topography, the geothermal heat flux can vary a lot on short spatial scales, so the importance of this should be at least discussed.*

We now discuss the impacts of geothermal heat flux in our discussion section.

"Another uncertainty could be from the spatially uniform geothermal heat flux that we assume in the model, as it may have a great spatial variation due to the mountain topography (Lüthi and Funk, 2001)."

*p7,22 So, the water content is assumed constant throughout the temperate ice? This*

*is problematic and will obviously introduce some inaccuracies.*

It's true that a constant water content may bring uncertainties in our results. Further efforts of including the water content computation and drainage system are still under development. We now have add a sentence for this in the discussion section.

"In addition, we can also improve our model ability by linking the water content in the temperate ice layer to a physical thermo-hydrological process in the future."

*p8,3 "compare to"*

Corrected.

*p8,6 The omission of convergent flow is only one possible (and likely) explanation, but there might be others, e.g. basal motion. This statement should be made more carefully.*

It's correct that our explanation is one of many possible reasons. The underes-

[Figure]

Figure 2: Modeled ice velocities for experiments E-ref (blue line), E-W (red line), and E-WS (green line). The glacier widths in the zone bounded by the vertical dashed lines are uniformly increased by 450 m.

timation of the ice surface velocities may possibly result from the neglect of the convergent flow from the west branch and an enhanced basal sliding which is not captured by our model in the confluence area. To verify this hypothesis, we conduct two other experiments, E-W and E-WS. In E-W the glacier widths are increased by 450 m at km $5.8 - 7.3$ as a proxy of including the impact of the convergent flow from the west branch (Fig. 2). In E-WS, except for the same glacier width increase as in E-W, we also increase $\lambda_{\max}$ by 200% and decrease $m_{\max}$ by 60% for accelerating the basal sliding at km $5.8 - 7.3$ (Fig. 2). We can clearly find that while both factors have a non-negligible contribution to the model results, the basal sliding may play a bit more important role in the confluence area. This indicates a need of considering glacier flow branches and spatially variable sliding law parameters in real glacier modeling studies. See p9–line3-10.

*p8,7 Here you should qualify "the modeled basal sliding velocities", IIUC. The reality, again, could be that basal sliding is much higher there. This could be elaborated upon in the Discussion.*

Yes, we discuss the modeled basal sliding velocities here. As suggested by the reviewer, we have conducted two other experiments in which the glacier widths are increased and the sliding law parameters are spatially tuned in the confluence area (see the above response). Then we discussed the impacts of convergent effects of the west branch and basal sliding (see p9–line8-10).

*p8,8 "observed": this is confusing, as you talk about model results. Better say: "the model predicts"*

We now use "The model predicts a TIZ overlain by cold ice over a horizontal distance of km 1.1 – 6.5".

*p8,9 add space between "110m"*

Fixed.

*p8,10 "ice fluxes" (not "ice flows")*

Corrected.

*p8,13 More important than matching temperatures would be a discussion of the heat fluxes. While the measurements show constant fluxes below 50 m depth below the surface, the model shows zones of warming and cooling (bends in the temperature profile). It would be important to understand the reason for these excursions from a straight line, is the shape of this profile due to advection, dissipation, or due the temperature history?*
*Closer to the surface (above 50 m depth) the measured gradient is much higher, which might reflect the thermal properties of the firn in a steady state (lower conductivity k). Since ice conductivity is assumed everywhere in the model, this might explain the difference there (cf. Fig. 5 in Luthi and Funk (2001) for a theoretical temperature profile with firn).*

This is a good question. To account for the thermal properties of the firn as suggested by the reviewer, we lower the conductivity ($k = 0.17$ W m$^{-1}$ K$^{-1}$) of the surface layer in the accumulation zone (experiment E-FC). Compared with the reference experiment (E-ref), E-FC results in higher temperature gradient above 60 m depth (Fig. 3a). Nevertheless, this cannot explain the deviation of the modeled temperature profile from a nearly straight line below 30 m depth. We also conduct other experiments to investigate the possible factors affecting the shape of the modeled temperature profile by adjusting the parameters, i.e., ELA, firn temperature and horizontal grid resolution. We find that the bend of the modeled temperature profile at the borehole is strongly influenced by the discontinuous thermal surface boundary condition accross the accumulation and ablation zones. The borehole is

located in the upper ablation area (4971 m a.s.l.), and is close to the snow line (around 4980 m a.s.l.). Therefore, the modeled temperature at the borehole can be influenced by the horizontal advection of relatively warm ice due to released latent heat from the accumulation zone. The higher temperature gradient in the upper part of the modeled profile demonstrates the impacts of horizontal heat advection from the upstream. We also compare the modeled temperature profiles below the borehole, which show little impacts from the upstream heat advection (Figure 3b). In 2011, we observed that the ice drilled below the depth of 166 m was wet, indi-

[Figure]

Figure 3: (a) Comparison of modeled temperature profiles at the borehole site. Blue line shows the modeled temperature in the reference experiment, while red line shows the result of experiment E-FC in which the firn conductivity is taken into account. (b) Comparison of modeled temperature profiles in the reference experiment. Blue line shows the modeled temperature at the deep borehole (4971 m a.s.l.). The red, green and purple lines show the modeled temperature profiles at 4954 m a.s.l., 4945 m a.s.l. and 4923 m a.s.l., respectively. Measured borehole temperatures are shown in dots. The pressure-melting point is shown by the dotted line.

cating the temperate ice layer there was possibly thicker than our model prediction (around 5.6 m). As our 2D flow-band model assumes a simple parameterization of the surface thermal boundary conditions, and neglects the convergent flow from the other cirques, it cannot capture the complex heat flow at the deep borehole site. In the future, we may perhaps try a 3D Stokes model and see if there would be something different.

*p8,27 It would be helpful to also show a graph of TIZ thickness (a second panel in Fig. 8c). It appears that the bed is temperate almost everywhere in the blue and*

*green model runs, but with very small TIZ.*

Good suggestions. We have shown the TIZ thicknesses in the double-Y-axis graphs, i.e. Fig. 10c, 11c and 12c. It's correct that a large region of the bed is temperate as predicted by the experiments E-ref and E-20m. Thick temperate basal ice appears in km 2 – 6, while temperate ice in other places is only one layer.

*p8,31 "above" (leave away "in")*

We now delete "in".

*p9,11 ff instead of "drop" and "remove" you could consistently use "neglect" or "leave away"*

Thanks. We now use "neglect" and "leave away".

*p9,25 qualify basal sliding by modeled*

Corrected.

*p9,28 leave away "higher-order"*

Changed.

*p10,2 "physically" should be "physical"*

Corrected.

*p10,9 consolidate the two citations*

Corrected.

*p10,11 Past changes can have a very important impact (see for example Luthi et al. (2015)), as are warming processes in the firn (e.g. Machguth et al. (2016))*

The corresponding sentences have been reformulated as "The assumption of steady state neglects the transient effects of past climate and glacier changes, which can have a very important impact on the shape of temperature profile (Lüthi et al., 2015; Gilbert et al., 2015).".

*p11,16 Replace "e.g." with "of" (these are not just examples, but an exhaustive list of measurements used in the study).*

Fixed.

*p11,21 No need to show the symbol "(u)" here again (leave away).*

We now delete "(u)".

*Fig 1 A nice overview photograph would help setting the scene for this remote glacier that most readers wont know.*

Good suggestion. We now use a Landsat 8 satellite image of LHG12 Glacier.

*Fig 2 same labels on the horizontal axis of Figs. 3 and 4 would ease of comparison.*
Fixed.

*Fig 8d Caption: modeled and measured (lines vs symbols) should be interchanged.*
Corrected.

**References**

Funk, M., Echelmeyer, K. A., and Iken, A.: Mechanisms of fast flow in Jakobshavn Isbræ, West Greenland: Part II. Modeling of englacial temperatures, Journal of Glaciology, 40, 569–585, 1994.

Gagliardini, O., Cohen, D., Råback, P., and Zwinger, T.: Finite-element modeling of subglacial cavities and related friction law, Journal of Geophysical Research, 112, F02 027, 2007.

Pattyn, F.: Transient glacier response with a higher-order numerical ice-flow model, Journal of Glaciology, 48, 467–477, 2002.

Zhang, T., Xiao, C., Colgan, W., Qin, X., Du, W., Sun, W., Liu, Y., and Ding, M.: Observed and modelled ice temperature and velocity along the main flowline of East Rongbuk Glacier, Qomolangma (Mount Everest), Himalaya, Journal of Glaciology, 59, 438–448, 2013.

---

## Referee Report (RR1)

**Re-Review: Wang et al, tc-2016-38**

Dear colleagues,

This is a short re-review of the manuscript. The authors did a very good job addressing the points raised by the reviewers. In addition, they performed several supplementary model runs to further investigate model sensitivity. In my opinion this is now a nice, comprehensive study of the thermal state of a glacier in High Mountain Asia.

My recommendation is to publish the manuscript after some very minor comments have been addressed.

Sincerely, Martin Lüthi

**Specific comments**

p7, 21  say "61 grid points" or "61 grid cells"

p7, 25  you could add that "Details are given in Zhang et al. (2013)".

p7, 28  change "described in above" to "described above"

---

## Referee Report (RR2)

**Report on revised manuscript *"An investigation of the thermo-mechanical features of Laohugou Glacier No.12 in Mt. Qilian Shan, western China, using a two-dimensional first-order flow-band ice flow model"* by Wang et al.**

The authors have not made any change in this revised version of the manuscript concerning the way they constrain their steady state temperature field. I still think that it do not make any sense to constrain steady state temperature on one surface (20m depth) measurement which is obviously not representative of the steady state condition (or the author have to show that the climate has been unchanged in the last 50 years in the region). It also makes no sense to try to fit the upper part (above 50 m depth) of their deep temperature profile with the modeled steady state temperature profile. The high temperature gradient in the upper part of this profile is very likely due transient surface cooling in response to reducing firn area. That would mean that results clearly underestimate the amount of temperate ice using the steady state assumption that way. A better approach would be to try to fit the bottom part of the temperature profile, it doesn't matter if the steady profile fit the upper part or not, there is no reason for that...

Consequently, I still don't think the current version of the manuscript deserves publication in The Cryosphere.

**I suggest the author to:**

- First, calibrate a steady state temperature field by trying to match (closely) the bottom part of their temperature profile (probably using an ELA more representative of a steady state for the glacier mass balance rather than using the ELA of one particular year!!)

- Then, if no air temperature time series are available, the author should try different past air temperature scenarios in order to get transient temperature field in accordance with the englacial temperature measurement they have. The transient scenario have to include both transient surface temperature and ELA evolution. Also I suggest the author to look if some reanalysis product of air temperature are available in the region for constrain the transient model.

**General comments:**

- Sorry if I was unclear but I suggested to the author to use in the ablation area a parametrization that link Tair to Tsbc (Tsbc = Tair +c) for performing transient simulation. What does the use of

such parametrization bring now in this revised manuscript??? This is not improving the way that boundary condition are addressed. Figure 7 can be deleted.

- Author should not use one particular year of ELA (2011 here) for modeling a steady state temperature but should use the mean ELA over the last 50 years or at least something close to the steady mass balance ELA… This lead also to wrong surface boundary condition.

---

## Referee Report (RR3)

**Report on revised manuscript "*An investigation of the thermo-mechanical features of Laohugou Glacier No.12 in Mt. Qilian Shan, western China, using a two-dimensional first-order flow-band ice flow model*" by Wang et al.**

I congratulate the authors for the effort they made to take into account transient climate effect on the thermal regime. This significantly improves the manuscript quality. Everything is now present in the paper but one step is missing: the author have now to use their transient model to calibrate correctly the diagnostic run. Again, it makes no sense to fit the entire measured **UNSTEADY** temperature profile with a diagnostic run. The authors even nicely show in the paper that the profile is affected by a strong surface cooling between 1970 and 2011. So, as I describe in the "General comment" the correct approach would be to fit the measured deep borehole with the transient model only. The diagnostic run has to be calibrated as **an initial condition of the transient run**.

I think that the author made a great effort to improve the paper but one crucial point is still not addressed and this require to be done before publication in The Cryosphere. The model is now complete and this should not be too hard. This is just concerning the way the surface parameter of the diagnostic run are calibrated. This is important because the presented modeled thermal regime depend strongly on this. The current diagnostic run likely underestimates the amount of temperate ice because of a too high ELA.

**General comments**

The authors do not specify the initial condition in the transient run which is crucial for such short simulation. Even with the 10 years spin-up, results are still dependent of the initial condition for a glacier of this size. It would make more sense to initialize the transient simulation from a diagnostic run.

The authors still calibrate the diagnostic run by fitting the complete measured deep profile. Again, this is not the good approach! Your work on mass balance and ELA reconstruction show a significant change in ELA elevation leading to significant cooling in the deep borehole during the recent years. This cooling very likely did not reach steady state yet, you cannot constrain your diagnostic run on this. However, you have now everything to do it properly:

Find (ELA, Tdep, c)$_{steady}$ in a way that your model fits the measurement **AFTER** running the transient model using the diagnostic run as initial condition.

So :

1- Run the diagnostic model with (ELA,Tdep,c)$_{steady}$
2- Run the transient model until 2011 with the diagnostic model as an initial condition
3- Compare the result of the transient model with measurement to adjust (ELA,Tdep,c)$_{steady}$

**Specific comments**

Line 17 p7 – I would remove "result in lower velocity values (Sugiyama et al., 2014)", this is an indirect consequence of the cold bias. Better to focus on temperature only here

Line 26 p7 – Use other symbol ($c$ is already used in eq. 15)

Line 11 p13 – "The transient model appears to bring more heat from the accumulation basin to downtream over the past decades." Unclear plus typo (downstream). Also, this is because of the 10 years spin up with the 1960-1970 climate condition that give wider accumulation area… this not link to the transient climate that actually tend to cool the glacier…

---

## Referee Report (RR4)

The manuscript "An investigation the thermo-mechanical features of Laohugou Glacier No. 12 in Mt. Qilian Shan, western China, using a two-dimensional first order flowband ice flow model" has already gone through one or two rounds of reviews and the authors have carefully addressed the previous comments. I will thus keep my review rather short to avoid pushing the manuscript into a new direction, making it a never ending story. I like the figures.

Andy Aschwanden, University of Alaska Fairbanks

**General comments**

- My main concern is the use of $n = 3$ as the exponent of both the flow law (Eq. 2) and as the exponent in the sliding law (Eq. 13). While there is relatively good evidence that supports $n = 3$ in the flow law, this is not the case for the sliding law. Simulated flow speeds are quite sensitive to the choice of $n$. For a Greenland example, see Supplementary Fig. 1 in Aschwanden et al. (2016). In my experience this is one of the most crucial parameters in ice dynamics so the authors would need to convince me that their conclusions remain robust.

**Specific comments**

**p. 1, l. 19**  as well as later. "...of approximately $50.5\,\mathrm{km}^2$". In combination with the word "approximately", I would only reduce the number of significant digits to "approximately $50\,\mathrm{km}^2$.

**p. 2, l. 15**  It seems appropriate to cite the oldest relevant manuscript. The importance of the surface thermal boundary has been discussed by Blatter and Kappenberger (1988). There is also a bunch of older literature on Storglaciären, northern Sweden that talks about the refreezing of snow melt in the firn area. My memory is a bit more shaky there, I may cite some papers in Aschwanden and Blatter (2005).

**p. 2, l. 28**  change "basal slip" to "basal motion" so you can avoid talking about the actual mechanism by which the motion occurs.

**p. 3, l. 7**  change "mainstream" to "main branch"

**p. 3, l. 27**  change "is an order" to "is on the order"

**p. 5, l. 9**  The first order model was first described by Blatter (1995) but Pattyn (2002) did a nicer job explaining the equations. I suggest using both citations.

**p. 6, l. 16**  . . . in the temperate zone refreezes

**p. 7, l. 28**  change "We denote" to "We use"

**p. 9, l. 8–9**  Are you really using a "zero surface mass balance" for the relaxation? Maybe I mis-interpret this, but to me, this means that the surface mass balance is zero at every grid point at the surface. That would be an odd choice. Commonly, one uses a constant-in-time surface mass.

**p. 10–11, Parameter sensitivity**  I am not so familiar with the temperature model used here but it appears that the authors try to simulate a polythermal with non-polythermal physics, which may question some of the conclusions. Please correct me if I have misread the manuscript. For example, the authors state that "a larger geothermal heat flux can result in larger TIZs but has limited impact on modeling ice velocity" (I think it should read: "a larger geothermal heat flux can result in a larger TIZ but has a limited impact on simulated ice velocities."). This is not a surprise as the model used here does not soften the ice as a function of the water content. Nonetheless, I agree with the conclusion. Using a polythermal model that includes this feedback does not change the shape of the TIZ and the simulated velocities much (viscosity is a linear function of water content). This has been discussed by Greve (1997a,b); Aschwanden et al. (2012) and recently by Hewitt and Schoof (2017).

**p. 12., l. 29–31**  This needs to be reformulated. From Fig. 12a, I cannot see why the temperature profile is determined by the thermal boundary condition and by the ELA. "can be remarkably determined" is not proper English. Also it needs to read ". . . , while the ELA controls. . . "

**p. 14, l. 18**  change "which "cools down" the ice temperature to "which "cools down" the ice. One can warm and cool ice, but temperature can only be lowered or raised.

**Figure 1a**  The map looks nice but please state what the background image is. Also, it's pretty dark so it is hard to see any features. Maybe use a pan-sharpended RGB composite instead?

**References**

Aschwanden, A. and H. Blatter (2005). Meltwater production due to strain heating in Storglaciären, Sweden. *J. Geophys. Res.*, **110**(F4). doi:10.1029/2005JF000328. URL http://www.agu.org/pubs/crossref/2005/2005JF000328.shtml.

Aschwanden, A., E. Bueler, C. Khroulev, and H. Blatter (2012). An enthalpy formulation for glaciers and ice sheets. *J. Glaciol.*, **58**(209), 441–457. doi:10.3189/2012JoG11J088.

Aschwanden, A., M. A. Fahnestock, and M. Truffer (2016). Complex Greenland outlet glacier flow captured. *Nature Communications*, **7**, 10524. doi:10.1038/ncomms10524. URL http://www.nature.com/doifinder/10.1038/ncomms10524.

Blatter, H. (1995). Velocity and stress fields in grounded glaciers: a simple algorithm for including deviatoric stress gradients. *J. Glaciol.*, **41**, 333–344.

Blatter, H. and G. Kappenberger (1988). Mass balance and thermal regime of Laika ice cap, Coburg Island, N.W.T., Canada. *J. Glaciol.*, **34**(116), 102–110.

Greve, R. (1997a). A continuum-mechanical formulation for shallow polythermal ice sheets. *Phil. Trans. R. Soc. A*, **355**(1726), 921–974. doi:10.1098/rsta.1997.0050. URL http://rsta.royalsocietypublishing.org/cgi/doi/10.1098/rsta.1997.0050.

Greve, R. (1997b). Application of a Polythermal Three-Dimensional Ice Sheet Model to the Greenland Ice Sheet: Response to Steady-State and Transient Climate Scenarios. *J. Climate*, **10**, 901–918.

Hewitt, I. J. and C. Schoof (2017). Models for polythermal ice sheets and glaciers. *The Cryosphere*, **11**(1), 541–551. doi:10.5194/tc-11-541-2017. URL http://www.the-cryosphere.net/11/541/2017/https://www.the-cryosphere.net/11/541/2017/.

Pattyn, F. (2002). Transient glacier response with a higher-order numerical ice-flow model. *J. Glaciol.*, **48**(162), 467–477. doi:10.3189/172756502781831278.

---

## Author Response (AR2)

**Response to Referee Martin Lüthi**

**Yuzhe Wang**

We thank the reviewer very much for the positive and encouraging comments of our manuscript.

**Specific comments**

*p7, 21 say 61 grid points or "61 grid cells"*
Changed.

*p7, 25 you could add that "Details are given in Zhang et al. (2013)".*
Included. Thanks.

*p7, 28 change "described in above" to "described above"*
Fixed.

**Response to Anonymous Referee #1**

Yuzhe Wang

We are grateful to the reviewer for providing insightful comments and contructive suggestions, which substantially improved the manuscript. Our response to all the comments is given below.

**Suggestions**

*First, calibrate a steady state temperature field by trying to match (closely) the bottom part of their temperature profile (probably using an ELA more representative of a steady state for the glacier mass balance rather than using the ELA of one particular year!!)*

Agreed. For the dianostic simulations, we tested many combinations of parameters in the surface thermal boundary condition (i.e. ELA, $T_{\mathrm{dep}}$, and $c$). The performance of each model run was evaluated by the root-mean-squres (RMS) of the differences between the lower part (below 40m deep) of measured and modeled temperatures in the deep borehole (Fig. 1). By this sensitivity experiment, we selected ELA = 4990 m a.s.l., $T_{\mathrm{dep}} = -2.1\,^{\circ}\mathrm{C}$, and $c = 4\,^{\circ}\mathrm{C}$.

Fig. 2 shows the diagnostically modeled velocity and temperature fields. Measured and modeled borehole temperature profiles are in very good agreement (Fig. 2d). Modeled surface velocities also fit well with the observations (Fig. 2e). It should be noted that we used a relaxed free suface for diagnostic simulation (see more details in our revised manuscipt).

*Then, if no air temperature time series are available, the author should try different past air temperature scenarios in order to get transient temperature field in accordance with the englacial temperature measurement they have. The transient scenario have to include both transient surface temperature and ELA evolution. Also I suggest the author to look if some reanalysis product of air temperature are available in the region for constrain the transient model.*

In this version, we've reconstructed the daily surface air temperatures on the studied glacier using the air temperature data from surrounding meteorological stations. We also collected gridded precipitation datasets covering the Qilian Shan. The downscaled meteorological data was used to force a surface mass balance model to determine the ELA of the glacier. More details about the data processing and the model improvements were included in the revised

manuscipt and the supplement.

**Comments**

*Sorry if I was unclear but I suggested to the author to use in the ablation area a parametrization that link Tair to Tsbc (Tsbc = Tair + c) for performing transient simulation. What does the use of such parametrization bring now in this revised manuscript??? This is not improving the way that boundary condition are addressed. Figure 7 can be deleted.*

Thanks for clarifying the parameterization. In this revised version, we calibrated the parameter $c$ by fitting the lower part (below 40m deep) of the temperature profile between the modeled and the measured (see our first response).

*Author should not use one particular year of ELA (2011 here) for modeling a steady state temperature but should use the mean ELA over the last 50 years or at least something close to the steady mass balance ELA... This lead also to wrong surface boundary condition.*
Agreed. Please see our first response.

[Figure]

Figure 1: Root mean squares (RMS) of differences between measured and modeled temperature profiles in the deep borehole. The red circle indicates the minimum of RMS. The parameter $T_{\mathrm{dep}}$ is varied from $-3.3\,°\mathrm{C}$ to $-1.5\,°\mathrm{C}$ with a step-size of $0.3\,°\mathrm{
[revised manuscript text omitted]

**5.2 Diagnostic simulations**

In our diagnostic simulations, we assume a thermal steady-state  condition and use the relaxed present-day geometry of LHG12. First, we explore the model sensitivities to geometrical bed parameters ($\lambda_{\max}$ and $m_{\max}$), ratio of basal effective

pressure ($\phi$), water content ($\omega\mu$), geothermal heat flux ($G$), and the valley shape index ($b$), and . Next, we tune the surface thermal boundary parameter condition parameters, i.e., ELA, $T_{\text{dep}}$ and $c$ . We then inspect three to fit the modeled steady-state temperature profile to observations in the deep borehole (Fig. 1). We then investigate the modeled horizontal velocity field and thermal structure of LHG12, and compare the diagnostic simulation results with observations. We also perform experiments to investigate sensitivity of the thermomechanical model to different surface thermal boundary conditions (E-ref and E-air). Finally, E-air, and E-20m)by comparing their model outputs with *in situ* ice temperature observations in the deep borehole at site 3. In addition, we perform four experiments (E-advZ, E-advX, E-strain, and E-slip) to investigate the impacts explore the effects of heat advection, strain heating, and basal sliding on the thermal field and flow characteristics distribution and flow dynamic behaviours of LHG12.

**5.3 Parameter sensitivityTransient simulations**

To investigate the impacts of historical climate conditions on the thermal regime of LHG12, time-dependent simulations are performed from 1961 to 2011. In our simulations, we assume that the surface temperature ($T_{\text{dep}}$) in the accumulation zone is constant in time and space. Due to a lack of in-situ observations (e.g. firn thickness, firn densities) and coupled heat-water transfer model (e.g. Gilbert, 2012; Wilson, 2013), we do not simulate the complex processes of heat exchange in the accumulation zone. To understand how the thermal status varies over time in the deep borehole, we design three (cold, warm and reference) transient simulations by setting $T_{\text{dep}}$ to $-5\,°C$, $-1\,°C$ and $-2.1\,°C$ (as calibrated in our steady state simulations; see Sect. 6.1.1), respectively.

In the transient model, the ice surface temperature ($T_{\text{sbc}}$) and ELA are allowed to vary in time. However, we keep the glacier geometry fixed in the transient simulations for the following two reasons: 1) The tributaries of LHG12, which our flowband model neglects, may have non-negligible inflow ice fluxes that impact the mass continuity equation; 2) The mean surface elevation change above 4600 m a.s.l. (the confluence area) over 1957–2014 is close to negligible (approximately $-10.4$ m, around 4.4% of the ice thickness) (See Fig. S10 in the Supplement). Before we run the transient experiments, we perform a 10-year spin-up using the mean values of surface air temperatures and ELAs during 1961–1970.

**6 Simulation results and discussions**

**6.1 Diagnostic simulations**

**6.1.1 Parameter sensitivity**

As shown in Figs. 5 and 6, we conduct a series of sensitivity experiments to investigate the relative importance of different model parameters ($\lambda_{\text{max}}$, $m_{\text{max}}$, $\phi$, $\omega\mu$, $G$, $b$, $c$) on ice flow speeds and temperate ice zone (TIZ) sizes by varying the value of one parameter while holding the other parameters fixed. others fixed. We set ELA to 4980 m a.s.l., as observed in the 2011 remote sensing image of LHG12. $T_{\text{dep}}$ and $c$ are set to -2.7 °C and -4.3 °C, which are calculated from the 20 m deep ice temperatures and mean annual air temperatures for the two shallow boreholes in the accumulaiton basin, respectively.

[revised manuscript text omitted]
  $T_{\text{dep}} = -5^{\circ}\text{C}$ (cold case),  $-2.1^{\circ}\text{C}$ (reference case) and  $-1^{\circ}\text{C}$ (warm case). ()  (b)  The variations of ELA and summer air temperature at 4200 m a.3 during 1961–2011. (c)  The vertically averaged ice velocity profiles along $x$ for the diagnostic and  transient simulation (reference case); (d)  The vertically averaged ice temperature profiles along $x$ for the diagnostic and transient simulation (reference case);

---

## Author Response (AR3)

[revised manuscript text omitted]
10 ~~verify the first two hypotheses by conducting two other experiments, E-W and E-WS. In E-W the glacier widths are increased by 450 m at km $5.8 - 7.3$ as a proxy of including the impact of the convergent flow from the west branch (Fig. 10). In E-WS, except for the same glacier width increasing as in E-W, we also increase $\lambda_{\mathrm{max}}$ by 100and decrease $m_{\mathrm{max}}$ by 33for accelerating the basal sliding at km $5.8 - 7.3$ (Fig. 10). We can clearly find that while both factors have a non-negligible contribution to the model results, the basal sliding may play a bit more important role in the confluence area. This indicates a need of considering~~
15

~~The model predicts a TIZ overlain by cold ice over a horizontal distance of km $0.6 - 7.4$ (Fig. **??**c). In addition, we further compare our model results with the *in situ* 110 m deep ice temperature measurements (Fig. **??**d). Modeled and measured borehole temperature profiles show a close match within a root-mean-square difference of 0.3 K below the 10 m depth. Because *in situ* ice temperature data from below 110 m have not been obtained, we were unable to compare the modeled and measured~~
20 ~~ice temperatures at the ice-bedrock interface. Glacier thermal regime is largely influenced by the climate history and firn thickness . However, our diagnostic simulation assumes a thermal steady-state and applies a simple thermal boundary condition. The good agreement between modeled and measured temperature profiles is likely due that we selecte a snow line altitude representing a steady state for the glacier mass balance, and the surface thermal boundary condition includes the effect of refreezing meltwater in the accumulation zone.~~

[revised manuscript text omitted]
_{\mathrm{dep}}$ is varied from $-3.3\,^{\circ}\mathrm{C}$ to $-1.5\,^{\circ}\mathrm{C}$ with a step-size of $0.3\,^{\circ}\mathrm{C}$, while $c$ is varied in the range of  $1$–$6\,^{\circ}\mathrm{C}$ with a  step size of $1\,^{\circ}\mathrm{C}$. The equilibrium line altitude (ELA) is fixed in each panel.

[Figure]

**Figure 8.** Modeled ice temperatures, velocities and TIZ thicknesses along the CL for diagnostic experiments E-ref-D (black solid line), E-air (gray solid line), E-advZ (black dash-dotted line), E-advX (gray dash-dotted line), E-strain (black dashed line), and E-slip (gray dashed line). (a) Modeled column mean ice temperatures. (b) Modeled basal ice temperatures. (c) Modeled surface horizontal velocities. (d) Modeled TIZ thickness.

[Figure]

**Figure 9.** Comparison of measured and transiently modeled horizontal velocities and ice temperatures of LHG12. (a) Modeled distribution of horizontal ice velocity. (b) Measured (markers) and modeled surface (solid line) and basal (dashed line) horizontal velocities along the CL. (c) Modeled distribution of ice temperature. The blue dashed line indicates the CTS position, and the black bar indicates the location of the deep ice borehole. (d) Measured (dots) and modeled (solid line) ice temperature profiles in the deep borehole. Pressure-melting point is indicated by the dotted line.

[Figure]

**Figure 10.** Modeled  surface (black lines) and basal (gray lines) horizontal velocities along the CL for experiments  E-ref-T ( solid line), E-W ( dotted line), and E-WS ( dash-dotted line). The glacier widths in the zone of km 5.8 – 7.3 (bounded by the vertical dashed lines) are increased by 450 m for E-W and E-WS. In E-WS, we also include a basal sliding enhancement between km 5.8 – 7.3.

[Figure]

**Figure 11.** (a) Modeled temperature profiles in the deep borehole for experiments E-ref-T (black line), E-cold (blue line), E-warm (red line), and E-ref-D (gray line). Dots indicate the measured ice temperature profile. Pressure-melting point is indicated by the dotted line. (b) The variations of ELA and summer air temperature at 4200 m a.s.l. during 1961–2011.

[Figure]

**Figure 12.** Differences of the column mean temperatures along the CL between the sensitivity experiments (E-high and E-low) and the reference experiment (E-ref-T). Gray lines indicate the initial temperature differences, whereas black lines indicate the temperature differences in 2011.

**Response to Anonymous Referee #1**

Yuzhe Wang, Tong Zhang

We are grateful to the reviewer for providing insightful comments and constructive suggestions, which substantially improved our manuscript and helped us develop numerical simulations of the glacier thermal regime. Our response to all the comments is given below.

**General comments**

*The authors do not specify the initial condition in the transient run which is crucial for such short simulation. Even with the 10 years spin-up, results are still dependent of the initial condition for a glacier of this size. It would make more sense to initialize the transient simulation from a diagnostic run.*

*The authors still calibrate the diagnostic run by fitting the complete measured deep profile. Again, this is not the good approach! Your work on mass balance and ELA reconstruction show a significant change in ELA elevation leading to significant cooling in the deep borehole during the recent years. This cooling very likely did not reach steady state yet, you cannot constrain your diagnostic run on this. However, you have now everything to do it properly:*

*Find $(ELA, Tdep, c)_{steady}$ in a way that your model fits the measurement AFTER running the transient model using the diagnostic run as initial condition.*

*So:*

*1- Run the diagnostic model with $(ELA, Tdep, c)_{steady}$*
*2- Run the transient model until 2011 with the diagnostic model as an initial condition*
*3- Compare the result of the transient model with measurement to adjust $(ELA, Tdep, c)_{steady}$*

We thank the reviewer for giving us constructive advices, which helps us model the thermal regime of LHG12 in a proper way. Following the reviewer's suggestions, we calibrate the parameters in surface thermal boundary condition ($ELA_0$, $T_{dep}$ and $c$), where $ELA_0$ is the initial ELA, by first running a diagnostic simulation with a given set of parameters ($ELA_0$, $T_{dep}$ and $c$) and then performing a transient simulation using the diagnostic run as an initial condition with time-dependent ELA and surface air temperature. This numerical process of finding the optimal parameter set of $ELA_0$, $T_{dep}$ and $c$ is repeated until we find a good agreement between the modeled and measured deep borehole

temperatures in 2011.

In this version of the manuscript, the structure is re-organized. The subsection "Comparison with *in situ* observations" is moved under the section "Transient simulations". In the discussions of the transient simulations, we also investigated the evolution of the borehole temperature profile during 1971–2011, and the impacts of the initial $ELA_0$ on the glacier thermal regime. Please find more details in the manuscript.

**Specific comments**

*Line 17 p7  I would remove "result in lower velocity values (Sugiyama et al., 2014)", this is an indirect consequence of the cold bias. Better to focus on temperature only here.*
Agreed. This has been removed.

*Line 26 p7  Use other symbol (c is already used in eq. 15)*
Fixed. We now use the symbol $m_c$ to represent the daily accumulation. In order to maintain a consistent style, we represent the daily ablation as $m_a$.

*Line 11 p13  "The transient model appears to bring more heat from the accumulation basin to downtream over the past decades." Unclear plus typo (downstream). Also, this is because of the 10 years spin up with the 1960-1970 climate condition that give wider accumulation area. . . this not link to the transient climate that actually tend to cool the glacier. . .*
Thanks for pointing out the typo. In this version of the manuscript, we performed transient simulations initialized with diagnostic runs. We also added a subsection to discuss the impacts of the initial condition on the simulation of the thermal regime.

**Response to Referee Andy Aschwanden**

**Yuzhe Wang, Tong Zhang**

We are grateful to the reviewer for providing constructive comments and positive feedbacks, which helped to improve the manuscript. Our response to all the comments is given below.

**General comments**

*My main concern is the use of $n = 3$ as the exponent of both the flow law (Eq. 2) and as the exponent in the sliding law (Eq. 13). While there is relatively good evidence that supports $n = 3$ in the flow law, this is not the case for the sliding law. Simulated flow speeds are quite sensitive to the choice of $n$. For a Greenland example, see Supplementary Fig. 1 in Aschwanden et al. (2016). In my experience this is one of the most crucial parameters in ice dynamics so the authors would need to convince me that their conclusions remain robust.*

[Figure]

Figure 1: Sensitivity of the modeled surface horizontal velocities and sliding velocities to the exponent $n$ in the Coulomb friction law.

We investigate the sensitivity of the modeled surface horizontal velocities and sliding velocities to the exponent $n$ in the Coulomb friction law by varying it from 1 to 3 with a step size of 0.5 (Fig. 1). It can be seen that the modeled surface velocities for $n < 2.5$ are much larger than those of the reference experiment (i.e., $n = 3$) due to largely increased basal sliding velocities. The largest basal sliding velocity at the km 2.6 accounts for approximately 70% of the surface velocity for $n < 2.5$, which is likely unrealistic. The modeled surface velocities for $n = 2.5$ is close to the results of the reference experiment. But it also predicts a larger basal sliding velocities compared to the reference result with a maximum value of about 15 m $a^{-1}$ at the km 2.0. In this study, we prefer to use $n = 3$ due to the agreements between the modeled and measured surface velocities. This setting would not affect our main conclusions about the thermo-mechanical features of LHG12. However, we argue that a varying $n$ in the Coulomb friction law may be possible due to the changing basal shear stresses along the flowline (Cuffey and Paterson, 2010).

**Specific comments**

*p. 1, l. 19 as well as later. ". . . of approximately 50.5 km² ". In combination with the word approximately, I would only reduce the number of significant digits to approximately 50 km².*

Agreed. We now write "... with a total ice volume of approximately 51 km³". Following this way of expression, the numbers combining with the word "approximately" in other places are also changed.

*p. 2, l. 15 It seems appropriate to cite the oldest relevant manuscript. The importance of the surface thermal boundary has been discussed by Blatter and Kappenberger (1988). There is also a bunch of older literature on Storglaciären, northern Sweden that talks about the refreezing of snow melt in the firn area. My memory is a bit more shaky there, I may cite some papers in Aschwanden and Blatter (2005).*

Agreed. We now cite relevant studies of the thermal regimes of glaciers in Arctic (Müller, 1976; Blatter, 1987; Blatter and Kappenberger, 1988) and in China (Huang et al., 1982). They have discussed the variable energy inputs in glacier zones that determine the near-surface temperature. The sentence is now reformulated as "Different energy inputs at the glacier surface determine the near-surface ice temperature. In particular, the latent heat released by meltwater refreezing in the percolation zone can largely warm the near-surface area (e.g., Müller, 1976; Huang et al., 1982; Blatter, 1987; Blatter and Kappenberger, 1988; Gilbert et al., 2014b). The importance of surface thermal boundary condition in controlling the thermal regime of a glacier has been recognized by recent thermo-mechanical glacier model studies (e.g., Wilson and Flowers, 2013; Gilbert et al., 2014a; Meierbachtol et al., 2015).".

*p. 2, l. 28 change "basal slip" to "basal motion" so you can avoid talking about the actual mechanism by which the motion occurs.*

Changed.

*p. 3, l. 7 change "mainstream" to "main branch"*

Changed.

*p. 3, l. 27 change "is an order" to "is on the order"*

Fixed.

*p. 5, l. 9 The first order model was first described by Blatter (1995) but Pattyn (2002) did a nicer job explaining the equations. I suggest using both citations.*

Agreed. This citation has been added.

*p. 6, l. 16 ... in the temperate zone refreezes*

Changed.

*p. 7, l. 28 change "We denote" to "We use"*

Changed.

*p. 9, l. 89 Are you really using a "zero surface mass balance" for the relaxation? Maybe I mis-interpret this, but to me, this means that the surface mass balance is zero at every grid point at the surface. That would be an odd choice. Commonly, one uses a constant-in-time surface mass.*

[Figure]

Figure 2: Differences between the two relaxation experiments with and without present-day surface mass balance. (a) Differences of the ice thicknesses after the relaxation. (b) Differences of the vertically averaged horizontal velocities after the relaxation. (c) Differences of the vertically averaged ice temperatures after the relaxation.

It is correct that we used the zero surface mass balance at each grid point. We did this because there was not enough measurements of surface mass balance covering the whole flowline. Now we perform a relaxation experiment using

the mean surface mass balance as modeled during 2009–2013. Then we compare the differences between the two relaxation experiments with and without present-day surface mass balance (Fig. 2). It shows that the modeled mean ice thickness after the relaxation using present-day surface mass balance is lower than that of using zero mass balance by about 3 m (Fig. 2a). The modeled vertically averaged horizontal velocities and temperatures along the flowline show little differences (Fig. 2b and c). In this version of the manuscript, we relax the free-surface using a constant present-day surface mass balance as suggested by the reviewer.

*p. 10–11, Parameter sensitivity I am not so familiar with the temperature model used here but it appears that the authors try to simulate a polythermal with non-polythermal physics, which may question some of the conclusions. Please correct me if I have misread the manuscript. For example, the authors state that "a larger geothermal heat flux can result in larger TIZs but has limited impact on modeling ice velocity" (I think it should read: "a larger geothermal heat flux can result in a larger TIZ but has a limited impact on simulated ice velocities."). This is not a surprise as the model used here does not soften the ice as a function of the water content. Nonetheless, I agree with the conclusion. Using a polythermal model that includes this feedback does not change the shape of the TIZ and the simulated velocities much (viscosity is a linear function of water content). This has been discussed by Greve (1997a,b); Aschwanden et al. (2012) and recently by Hewitt and Schoof (2017).*

We agree with this comment. Our model is currently not fully polythermal, as it doesn't include a mechanism that copes well with the water content in the temperate ice layer. We discuss it in the second paragraph of Section 6.1.1. Ideally, a warmer and larger TIZ may lead to softer basal ice, which may contribute more of the increase of the ice velocity field. To improve this, an englacial hydrology model should be coupled for calculating the water content and basal effective pressure. At this point, we are also trying to add the enthalpy scheme for the thermal model following Aschwanden et al. (2012), which we believe will eventually make some progress on this issue. We thank the reviewer for pointing it out.

*p. 12., l. 29–31 This needs to be reformulated. From Fig. 12a, I cannot see why the temperature profile is determined by the thermal boundary condition and by the ELA. "can be remarkably determined" is not proper English. Also it needs to read . . . , while the ELA controls. . ."*

In this version of the manuscript, this section has been largely changed. Please find more details in Section 6.2.2 of the revised manuscript.

*p. 14, l. 18 change "which "cools down" the ice temperature to "which cools down" the ice. One can warm and cool ice, but temperature can only be lowered or raised.*

Agreed. This has been fixed.

*Figure 1a The map looks nice but please state what the background image is.*

*Also, it's pretty dark so it is hard to see any features. Maybe use a pan-sharpended RGB composite instead?*

We now replace the black-white background image with a true-color Landsat 5 satellite image acquired on 22 September, 2011.

---

## Author Response (AR4)

**Response to Olivier Gagliardini**

**Yuzhe Wang and Tong Zhang**

Dear Dr. Gagliardini,
We are grateful for your suggestions and editorial work which have largely improved our manuscript. Following your suggestions, we have revised the manuscript. Our responses to the reviews are given below. The new version of our manuscript with changes is attached at the end of the document.

With respects,
Yuzhe

**Technical corrections**

*when citing a book, like Cuffey and Paterson (2010), would be nice to give the page number to easily find the information*
The page numbers have been added following the citation in a form such as "(Cuffey and Paterson, 2010, p. 60–61)".

*you are using K and °C alternatively during the manuscript. It would be easier to keep to °C all along?*
Agreed. We now have replaced the unit K with °C in our manuscript.

*page 13, line 24: and are not able to -¿ and is not able to*
Fixed.

*In FiG. 5 it would be nice to indicate what are the default values for all the varied parameters? Does it correspond to the green curve (which looks the same in all panels?)?*
We now have given the default values of the parameters investigated in the sensitivity experiments in Figs. 5 and 6.

[revised manuscript text omitted]